# MiRNAs shape mouse age-independent tissue adaptation to spaceflight via ECM and developmental pathways

Friederike Grandke [1,2,16], Shusruto Rishik[1,16], Viktoria Wagner [1,3], Annika Engel [1], Nicole Ludwig[4], Kruti Calcuttawala[3], Fabian Kern [1,2], Verena Keller[1,5], Marcin Krawczyk[5], Louis Stodieck[6], Virginia Ferguson [7], Amanda Roberts[8], Eckart Meese[4], Nicholas Schaum[3], Steven Quake[9,10,11], Tony Wyss-Coray [3,12,13,14] & Andreas Keller [1,2,15] ✉

As human space exploration accelerates, understanding the organism-wide molecular effects of longer spaceflight in mammals becomes increasingly critical. Non-coding RNAs like miRNAs are key to regulating this landscape. We thus analyze 686 small RNA samples of female mice from 13 solid organs at 3 and 8 months of age, after at least 3 weeks on the International Space Station and compare them to earth-bound controls. We observe significant spaceflight effects in systemic tissue remodeling pathways along the Fat-Liver-Pancreas axis and in heart, brain, spleen and thymus. The *MIR-17/92* and *MIR-1/133* families drive distinct molecular changes through specific gene targeting. Age-dependent changes, smaller in magnitude compared to age-independent changes, primarily involve tissue remodeling through *MIR-8*, *MIR-154* and *MIR-15* families in mesenteric adipose tissue, pancreas, and diaphragm. Our findings provide evidence on how spaceflight regulates mammalian gene expression in preparation for interplanetary spaceflight.

Since the 1960s, human presence in space has expanded, with more frequent missions and longer stays. With commercial costs to reach low earth orbit dropping by a factor of 20 in recent decades, from $54,500/kg to $2720/kg and continuing to drop[1], human activity and presence in space will keep increasing. The physiological impact of spaceflight on humans is substantial, with prolonged exposure leading to symptoms similar to degenerative diseases observed on Earth, such as muscle atrophy, bone loss, cardiovascular deconditioning and changes to the immune system[2,3]. These effects intensify with mission duration, particularly in extended spaceflights beyond Earth's orbit. Currently, expeditions to the International Space Station (ISS) are usually limited to a few weeks or months, but future missions to Mars and beyond will require even longer exposure to spaceflight stresses[4]. Molecular studies are needed to investigate spaceflight-induced physiological changes across tissues, aiming to identify biomarkers and therapeutic targets to mitigate adverse effects.

[1]Clinical Bioinformatics, Saarland University, Saarbrücken, Germany. [2]Helmholtz Institute for Pharmaceutical Research Saar (HIPS), Saarbrücken, Germany. [3]Department of Neurology and Neurological Sciences, Stanford University School of Medicine, Stanford, CA, USA. [4]Department of Human Genetics, Saarland University, Homburg, Germany. [5]Department of Medicine II, Saarland University Medical Center, Saarland University, Homburg, Germany. [6]BioServe Space Technologies, Aerospace Engineering Sciences, University of Colorado at Boulder, Boulder, CO, USA. [7]Department of Mechanical Engineering, University of Colorado at Boulder, Boulder, CO, USA. [8]Animal Models Core Facility, The Scripps Research Institute, La Jolla, CA, USA. [9]Chan Zuckerberg Biohub, San Francisco, CA, USA. [10]Department of Bioengineering, Stanford University, Stanford, CA, USA. [11]Department of Applied Physics, Stanford University, Stanford, CA, USA. [12]Wu Tsai Neurosciences Institute, Stanford University School of Medicine, Stanford, CA, USA. [13]Paul F. Glenn Center for the Biology of Aging, Stanford University, Stanford, CA, USA. [14]Stanford University, The Knight Initiative for Brain Resilience, Stanford, CA, USA. [15]PharmaScienceHub, Saarland University, Saarbrücken, Germany. [16]These authors contributed equally: Friederike Grandke, Shusruto Rishik. ✉e-mail: andreas.keller@ccb.uni-saarland.de

To better understand the molecular effects of spaceflight, numerous studies have investigated its impact on both humans and model systems. The NASA Twin Study[5] is one such effort examining the effects of spaceflight on humans, including changes in gene expression, DNA methylation, the microbiome and the immune system. The Space Omics and Medical Atlas (SOMA)[6] integrates this expanding data set, incorporating findings from the NASA Twins Study[5], JAXA CFE study[7,8], the SpaceX Inspiration4 crew[9–11], with data from the Axiom and Polaris missions. The dataset includes microbiome profiles, scRNA-seq, scATAC-seq, and bulk gene expression across multiple types of tissues and organisms.

Studying multiple modalities is important due to the interconnected regulatory nature of biological systems. One important source of regulation is through small RNAs, such as miRNAs. MiRNAs are 19–25-nucleotide RNA molecules that can bind to the 3' UTR of mRNAs and down-regulate them. Each miRNA can potentially target hundreds of such targets[12], with their sequences and often expression conserved between species such as *M. musculus* and *H. sapiens*[13,14]. Previously, researchers identified 13 miRNAs that respond to spaceflight effects in Murine solid tissue[15] and blood[16] through the cytokine *TGF-β1*. This highlights miRNAs' importance in regulating spaceflight effects. MiRNAs are also deregulated in tissues biopsied from chronic conditions like cardiovascular disease, cancer and aging[17–19]—conditions that share symptoms with the observed effects of long-term spaceflight[20–22]. Thus, we investigate whether the phenotypic similarity between degenerative disease and spaceflight stems from shared molecular mechanisms.

Human spaceflight studies solely rely on minimally invasive sampling, such as body fluids[6]. But degenerative conditions such as aging cause dysregulation across multiple tissue types[18]. A system-wide analysis of spaceflight exposure in humans has yet to be completed. Although multiple studies have examined the spaceflight-related deregulation of miRNA in individual murine tissues, none have analyzed multiple tissues from the same mice. Lastly, dysregulation in spaceflight arises not only due to microgravity and radiation but also from often overlooked factors such as housing and stress.

In this study, we analyzed miRNAs from 686 samples across 13 organs of young (3 months) and middle-aged (8 months) mice that were sent to the ISS (Flight). We compared them against mice living in standard conditions (Vivarium Ground Control) and mice living in an environment matched to ISS conditions (Habitat Ground Control). Furthermore, we compared miRNA profiling results with single-cell mRNA sequencing data from the same conditions (Supplementary Information) to identify miRNA-mRNA regulation. We also leveraged the Tabula Muris Senis (TMS) cohort to examine shared molecular pathways between aging and spaceflight.

## Results

### Spaceflight causes organ-specific murine small non-coding RNA expression changes

Various factors contribute to the physiological and environmental stress mice experience during spaceflight. Housing conditions, including the cage density, temperature, humidity, and carbon dioxide levels, can significantly influence the miRNA expression in cells derived from mouse tissues. To isolate the effects of spaceflight from the effects of housing across 13 tissues, we included HGC and VGC (Fig. 1a). Euthanizing mice at two time-points (matching timelines for controls and flight mice), one before returning to earth (TERM) and one after (LAR), allowed us to distinguish spaceflight-induced effects from the reentry-induced stress (Fig. 1b).

The 686 tissue samples split into 348 samples derived from the Live Animal Return (termed LAR) group and 338 samples derived from the ISS Terminated (termed TERM) group (Fig. 1c). In total, we sequenced 39.8 billion short reads with a median of 28.4 million per sample (interquartile range of 10.3 million), successfully mapping 70%

to the mouse genome (GRCm39) (Supplementary Data 1). Overall, gonadal adipose tissue (GAT), mesenteric adipose tissue (MAT), subcutaneous adipose tissue (SCAT) and brown adipose tissue (BAT) samples presented the highest percentage of mapped reads, while heart, lung, kidney and liver exhibited the highest proportion of miRNA-mapped reads (Fig. 1d). The majority of reads, however, mapped to tRNA (44%) and rRNA (14%), both crucial to mRNA translation (Fig. 1e).

Given this variation in RNA class distribution, we analyzed how much of the observed expression variance could be attributed to spaceflight. The RNA types that were most affected by spaceflight were piRNA fragments (median: 2.35), lncRNA fragments (median: 1.65) and tRNA fragments (median: 1.95). This confirms that several RNA classes respond to spaceflight, which supports previous findings about their role in spaceflight adaptation and astronauts' health[23,24]. For miRNAs, spaceflight effects accounted for a median of 1.85% of the variance in expression across tissues (Fig. 1f).

Since our small RNA sequencing technology is optimized for miRNA detection, this class of non-coding RNA provides the most reliable signal in our dataset. However, to ensure transparency and facilitate further research, we make the full dataset, including all detected RNA classes, available for other researchers (c.f. resource availability).

### Environment and spaceflight affect different tissues in murine miRNA profiles

After mirBase quantification and filtering for stably abundant miRNAs, we identified 1148 miRNAs in the LAR group and 1133 miRNAs in the TERM group (Fig. 2a). To visualize the full miRNA dataset, we projected all samples into a 2D UMAP space. Each organ forms a distinct cluster in the embedding (Fig. 2b) except for the adipose tissues (BAT, MAT, GAT, and SCAT), which cluster together. This reflects the dominant influence of the tissue type, which explains 53.1% of the variance in the data according to a Principal Variance Component Analysis (PVCA; Fig. 2c). In contrast, age, condition, and extraction method did not form distinct clusters in the embedding (Supplementary Fig. 1a–c). However, the extraction-tissue interaction and extraction itself contributed 2.5% and 0.8% of the variance. This underscores the impact of the extraction timepoint (TERM: day 21, LAR: day 43) and the preservation method (see Methods) on miRNA expression. Therefore, we focused our analysis on the LAR group, using the TERM group for cross-checking key findings.

During spaceflight, mice experience microgravity, increased radiation, and various environmental changes (Supplementary Fig. 1d), including altered housing conditions and dietary modifications. To separate spaceflight-induced changes from environmental effects, we included two ground control groups: one mimicking spaceflight conditions and one under standard laboratory conditions (see Methods). To identify the drivers of expression variation, we compared the proportion of miRNA variance attributed to spaceflight versus environmental factors. This analysis revealed an organ-specific effect: MAT, kidney and brain showed a greater proportion of variance due to housing conditions (Fig. 2d), whereas GAT, spleen, SCAT, thymus, liver and pancreas exhibited variance primarily driven by spaceflight. Among these, GAT, spleen and SCAT showed the strongest spaceflight-driven variance at 11.7%, 10.5%, and 8.9%, respectively.

The number of deregulated miRNAs further reinforced this pattern (Fig. 2e, Supplementary Data 2). GAT and SCAT ranked among the three tissues with the highest number of deregulated miRNAs. Specifically, GAT (66), heart (50), SCAT (33) and BAT (31) exhibited the highest number of deregulated miRNAs in spaceflight (FL vs. HGC), whereas diaphragm (1), kidney (2), MAT (2) and brain (7) had the fewest. Notably, although the kidney, MAT and brain had few deregulated miRNAs in the same comparison (FL vs. HGC), they presented significantly more deregulated miRNAs in FL vs. VGC and HGC

vs. VGC, suggesting a stronger influence of housing conditions on these organs. For example, the comparisons in the brain yielded 96 (FL vs. VGC) and 74 (HGC vs. VGC) deregulated miRNAs. To isolate spaceflight-related changes, we compared FL vs. HGC (reflecting spaceflight effects) and FL vs. VGC (reflecting both spaceflight and housing effects). Similarly, we assessed housing-related effects by comparing FL vs. VGC and HGC vs. VGC, with the latter isolating housing effects (Supplementary Fig. 1e). Correlation analysis revealed that fold-changes in FL vs. VGC and HGC vs. VGC match best in kidney, MAT and brain (Fig. 2f), suggesting that these organs were primarily influenced by housing conditions. In contrast, other organs exhibited stronger correlations between FL vs. HGC and FL vs. VGC, with GAT, spleen and SCAT among the most spaceflight-affected tissues, showing fold-change correlations above 0.7.

To refine our analysis at the level of individual miRNAs, we compared the deregulated miRNAs in the three comparisons. Diaphragm, thymus and liver showed the highest proportion of miRNAs commonly deregulated in FL vs. HGC and FL vs. VGC (diaphragm: 100%, thymus: 63%, liver: 55%). Notably, the diaphragm and thymus showed no overlap between FL vs. VGC and HGC vs. VGC (Fig. 3a, Supplementary Fig. 2). In contrast, the MAT and the kidney shared no deregulated miRNAs in the two flight comparisons but had a high proportion (MAT: 60%, kidney: 48%) shared between the housing comparisons. Since spaceflight-affected miRNAs should be deregulated in both FL vs. HGC and FL vs. VGC, we identified a total of 73 commonly deregulated miRNAs across all tissues that were consistently upregulated or downregulated in both comparisons. Of those 73 miRNAs, six belonged to the *MIR-8* family (miR-141-3p/5p,miR-200a/b/c-3p and miR-429-3p), which showed up in the heart and the pancreas and 5 belonged to the *MIR-17* family (miR-106a-5p, miR-20b-5p, miR-17-5p, miR-20a-5p and miR-18b-5p), showing up in the brain, GAT, heart and the thymus. Notably, 78% of these miRNAs presented tissue-specific deregulation, with the majority found in GAT (28 miRNAs), heart (11 miRNAs), liver (5 miRNAs) and SCAT (3 miRNAs) (Fig. 3b). Although no miRNA was deregulated across all tissues, 16 miRNAs were deregulated in multiple tissues, suggesting systemic spaceflight effects. In this analysis, members of the *MIR-8* family and the *MIR-196* family stand out, with miR-141-3p,

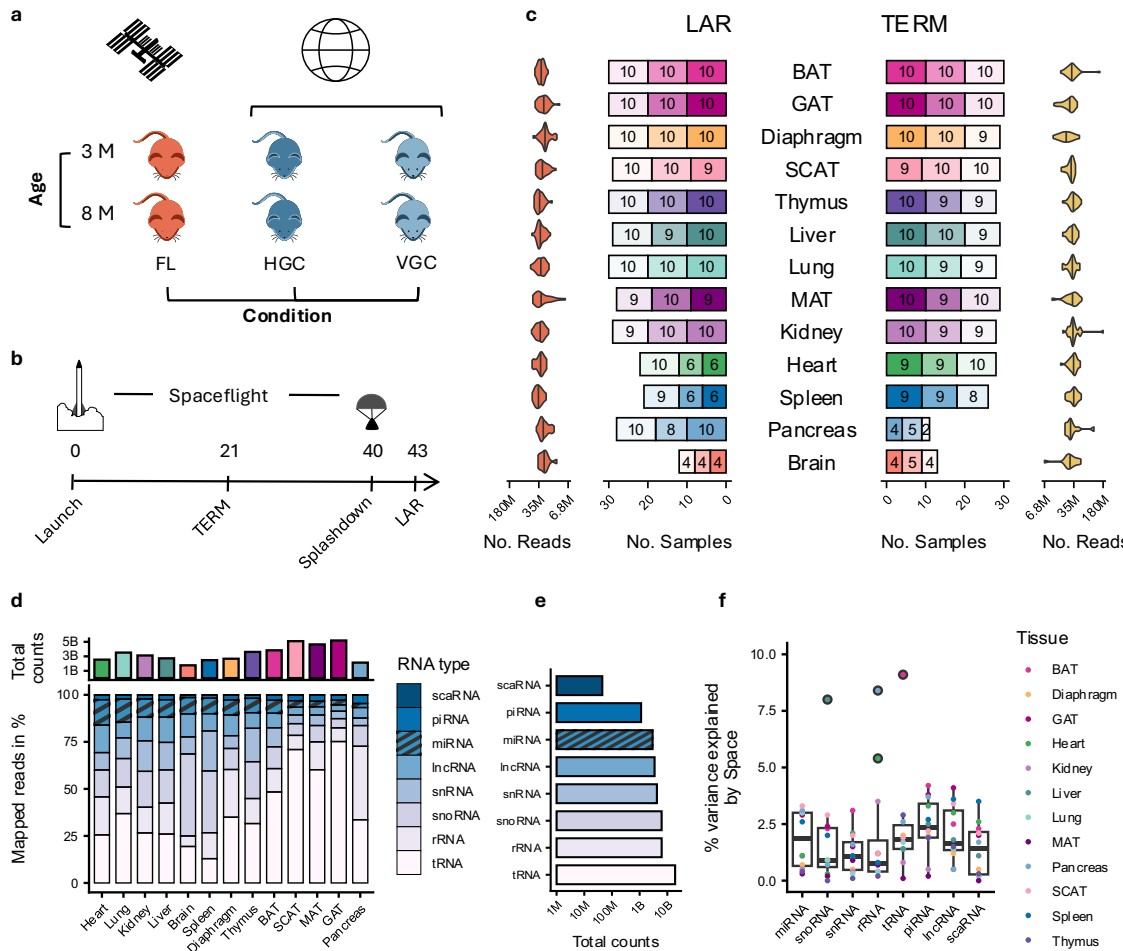

**Fig. 1 | Study dataset overview. a** We measured three mouse groups: Spaceflight (FL), Habitat Ground Control (HGC), mimicking housing conditions in flight and Vivarium Ground Control (VGC), representing typical mouse housing conditions. Each group splits into young (3 months) and old (8 months). **b** All experiments were carried out with 2 protocols: Live Animal Return (LAR), euthanized on Earth after 40 days of spaceflight and Immediate Termination (TERM), euthanized on the ISS after 21 days of spaceflight. **c** Sample sizes of FL, HGC, and VGC conditions across LAR and TERM and across tissues. Opacity indicates condition (FL: full opacity, HGC: medium opacity, VGC: light opacity). Violin plots show the distribution of the number of reads (log₁₀ transformed) in the small RNA sequencing library for each tissue for LAR and TERM. Marked lines in the violin plots are the median for each tissue. **d** Total counts of reads that aligned to the murine genome for each tissue, along with the percentage of reads for each tissue that aligned to the different non-coding RNA types. Source data are provided as a Source Data file. **e** Total number of reads (log₁₀ transformed) mapping to the different non-coding RNA types. Source data are provided as a Source Data file. **f** Distribution of the percentage of variance explained by Space condition in LAR for each Tissue by each type of non-coding RNA (number of biological replicates *n*: see **1c**). Source data are provided as a Source Data file. Boxplots show the median (center line), the first and third quartiles (box), and whiskers extending to the most extreme data points within 1.5 × IQR; values beyond this range are plotted as outliers.

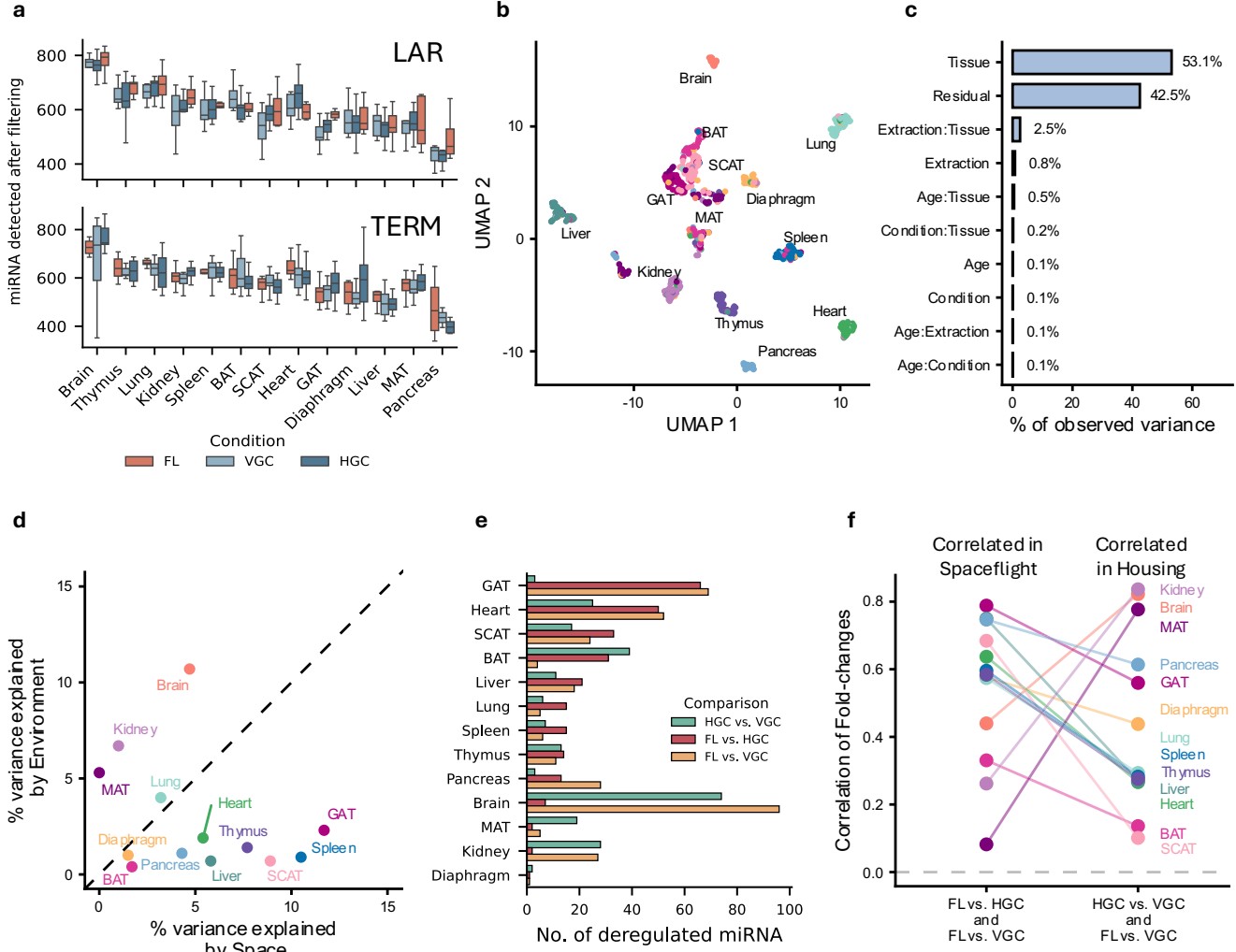

**Fig. 2 | Changes in miRNA profile after spaceflight. a** Distribution of miRNA detection counts based on Tissue and FL, HGC and VGC in LAR and TERM, respectively. A miRNA had to be detected at a count of five in at least 10% of each group to be considered as detected. (Number of biological replicates *n*: see Supplementary Data 1) Boxplots show the median (center line), the first and third quartiles (box), and whiskers extending to the most extreme data points within 1.5×IQR; values beyond this range are plotted as outliers. Source data are provided as a Source Data file. **b** Two-dimensional UMAP embedding of the 686 samples colored by tissue of origin. Source data are provided as a Source data file. **c** Percentage of variance explained by the variables tissue, extraction method, age, condition, their interactions and the Residual variance as calculated from a PVCA. **d** Scatterplot of % variance explained by environment versus % variance explained by Spaceflight for each tissue (PVCA). Environment corresponds to Normal Earth conditions (VGC) versus ISS or ISS-matched conditions (HGC and FL). Space effects mean whether the mice were exposed to Spaceflight (FL) or not (HGC and VGC). Source data are provided as a Source Data file. **e** Counts of the number of deregulated miRNA for each tissue and comparison. A miRNA was deemed to be deregulated if it had an absolute $\log_2$ fold-change greater than 1 and an absolute Cohen's D effect size >0.5 (see Methods). Source data are provided as a Source Data file. **f** Correlation of fold-changes in FL vs. HGC and FL vs. VGC, primarily capturing the effects of spaceflight, and in HGC vs. VGC and FL vs. VGC, primarily capturing the effects of housing. All correlations shown here were significant (adj. *P* value < 0.05, Pearson's correlation, BH adjustment). Source data are provided as a Source Data file.

miR-200a-3p, and miR-200b-3p affecting the heart and pancreas, and miR-196a-5p and miR-196b-5p affecting GAT and SCAT.

To elucidate the functions of these 16 systemic miRNAs, we conducted a pathway over-representation analysis (ORA with miEAA; Fig. 3c, Supplementary Data 3). The pathways clustered into three major groups: DNA synthesis/repair, developmental/structural changes and binding (Z-DNA and purines). The remaining pathways relate to development and cell structure. MiRNAs from the *MIR-1* family and miR-133a-3p influenced nearly all pathways, whereas others, like the *MIR-34* or *MIR-196*, primarily regulated developmental and structural processes. A broader pathway enrichment analysis relying on all miRNAs (based on Gene Set Enrichment Analysis; GSEA with miEAA) revealed that the most frequently enriched pathways occurred mainly in SCAT, spleen, and pancreas, with a lesser presence in GAT (Fig. 3d, Supplementary Data 4). Pathway clustering reaffirmed groups related

to development and cell structure. The remaining pathways correspond to extracellular matrix (ECM) and nuclear remodeling pathways.

To determine whether spaceflight effects persist after a shorter exposure and without the reentry-induced stress of the mice, we compared the LAR and TERM groups. Overall, the magnitude of deregulation for miRNAs was less pronounced in the TERM group (Supplementary Fig. 3a). Nevertheless, deregulation patterns correlated between the FL vs. VGC and FL vs. HGC in both the LAR (Pearson's correlation = 0.54, *p* value < 2.2e-16) and TERM group (Pearson's correlation = 0.58, *p* value < 2.2e-16) (Supplementary Fig. 3b). Filtering for miRNAs with consistent up- or downregulation in the LAR and TERM groups revealed that the pancreas (402 miRNAs), GAT (317 miRNAs) and lung (259 miRNAs) exhibited the highest overlap (Supplementary Fig. 3c). Twenty-two of those miRNAs were deregulated in one of the comparisons in LAR and in TERM (Supplementary Fig. 3d) and four

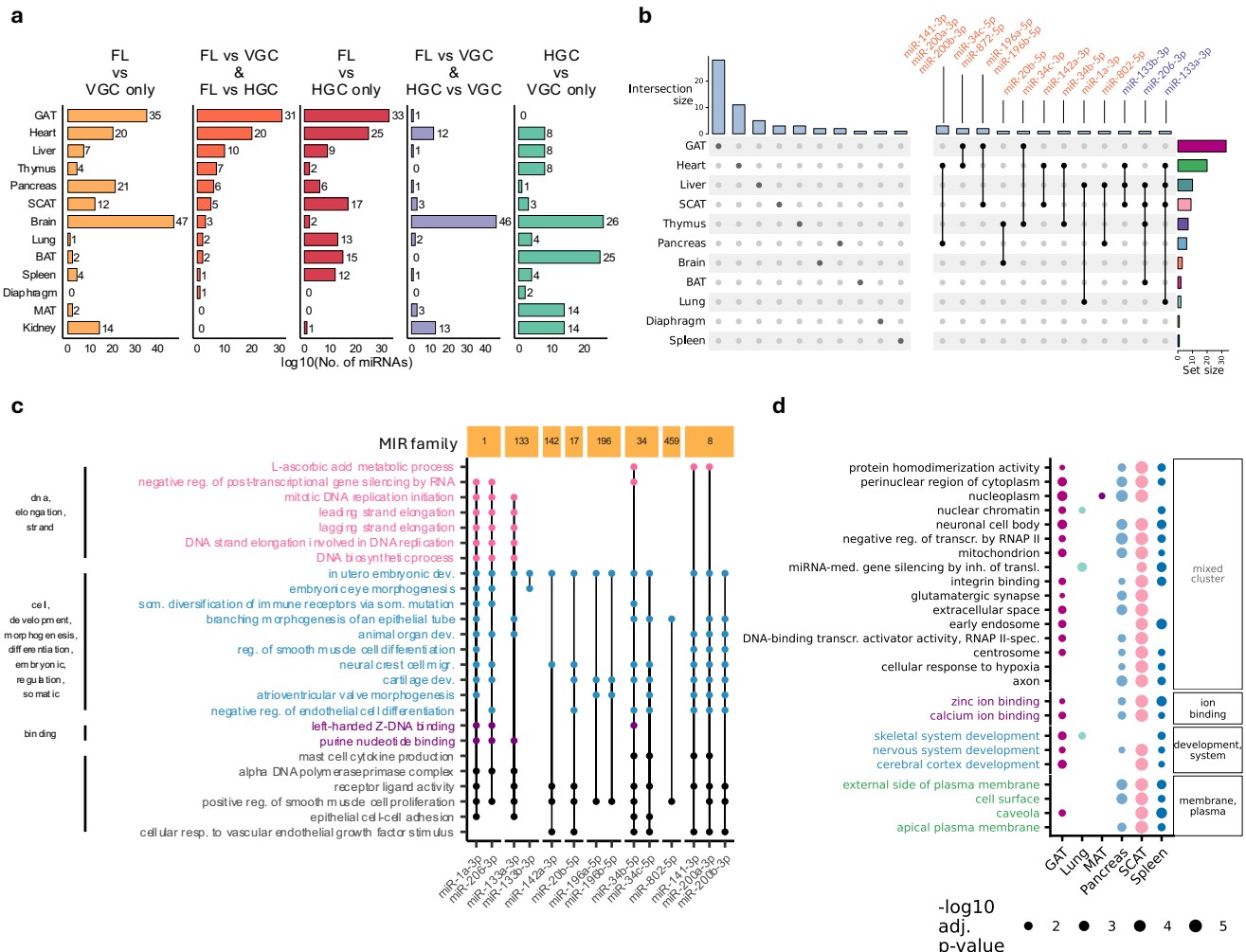

**Fig. 3 | Tissue-specific miRNA deregulation and pathway enrichment. a** Number of deregulated miRNA for each tissue's comparisons of conditions and the overlap of comparisons. The counts are log10-scaled. The tissues are sorted based on the subset "FL vs VGC & FL vs HGC". **b** Comparison between the spaceflight-related miRNAs within the different tissues. Spaceflight-related miRNA had to be deregulated in both FL vs. VGC and FL vs. HGC (Spaceflight effects) while also having the same direction of deregulation in both comparisons. MiRNAs that are deregulated in more than one tissue are shown to the right with their names written on top.

**c** Pathway analysis (ORA; miEAA; one-sided, BH adjusted; parameters see Methods) of the space-related miRNAs from (**b**) that occur in more than 1 tissue. The pathways are clustered based on similarity in GO and labeled with the most frequent words in the clusters (See method). **d** Top 25 pathways (GSEA; miEAA; one-sided, BH adjusted; parameters see Methods) that are significant in both FL vs. HGC and FL vs. VGC in most tissues. All pathways in the top 25 were enriched. Pathways for both (**c**, **d**) are clustered by similarity in GO, and pathway groups are reported as the most frequent words.

were consistently deregulated across all comparisons: miR-802-3p (liver), miR-200a-3p and miR-96-5p (pancreas) and miR-802-5p in (liver and pancreas). To link these changes to biological function, we performed a pathway analysis (using GSEA with miEAA) for TERM (Supplementary Fig. 4a). The top 25 pathways closely resembled those in LAR, particularly in binding-related processes, and modifications of the membrane and nuclear environment. Comparing the top 25 pathways in the LAR and TERM groups revealed that all were enriched in both conditions, though with tissue-specific differences. For example, the pancreas, GAT and lung showed similar pathway enrichment in LAR and TERM, whereas the spleen, thymus and diaphragm displayed more pronounced differences (Supplementary Fig. 4b). These similarities matched the overlaps found in pancreas, GAT, and lung, which exhibited comparable pathway activation, while the thymus and spleen showed condition- and tissue-specific patterns.

We furthermore compared the pathways found in LAR with the changes observed in the corresponding mRNA data (GSEA using Genetrail) (see Methods). The most frequently overlapping pathways in both datasets were enriched in cell types from SCAT, spleen and pancreas (Supplementary Fig. 4c). ECM organization and tissue-

remodeling dominated the affected pathways, particularly influencing endothelial cells in the kidney, epithelial cells from SCAT and immune-related cells such as macrophages in the spleen and kidney. These pathway similarities suggest a direct link between the deregulated mRNAs and the targeted genes of the deregulated miRNAs.

## Spaceflight causes systemic and tissue-specific miRNA deregulation in gene expression

The canonical function of miRNAs is to regulate gene expression by targeting mRNAs for degradation or translational repression. Having mRNA data (scRNA-seq) from the same experiment provides a unique opportunity to directly evaluate whether spaceflight-induced miRNA deregulation leads to corresponding gene-level changes. Since the mRNA dataset includes all 13 tissues, and LAR-Flight (Flight in mRNA) and LAR-HGC (Control in mRNA) share matched conditions, we identified miRNA targets that were also deregulated at the mRNA level, relying on tissue-level pseudo-bulk gene expression. This approach directly pinpointed deregulated miRNA-mRNA pairs (Supplementary Data 5). Of note, we did not filter for the direction of deregulation in mRNA and miRNA, as the miRNA-mRNA regulation does not always

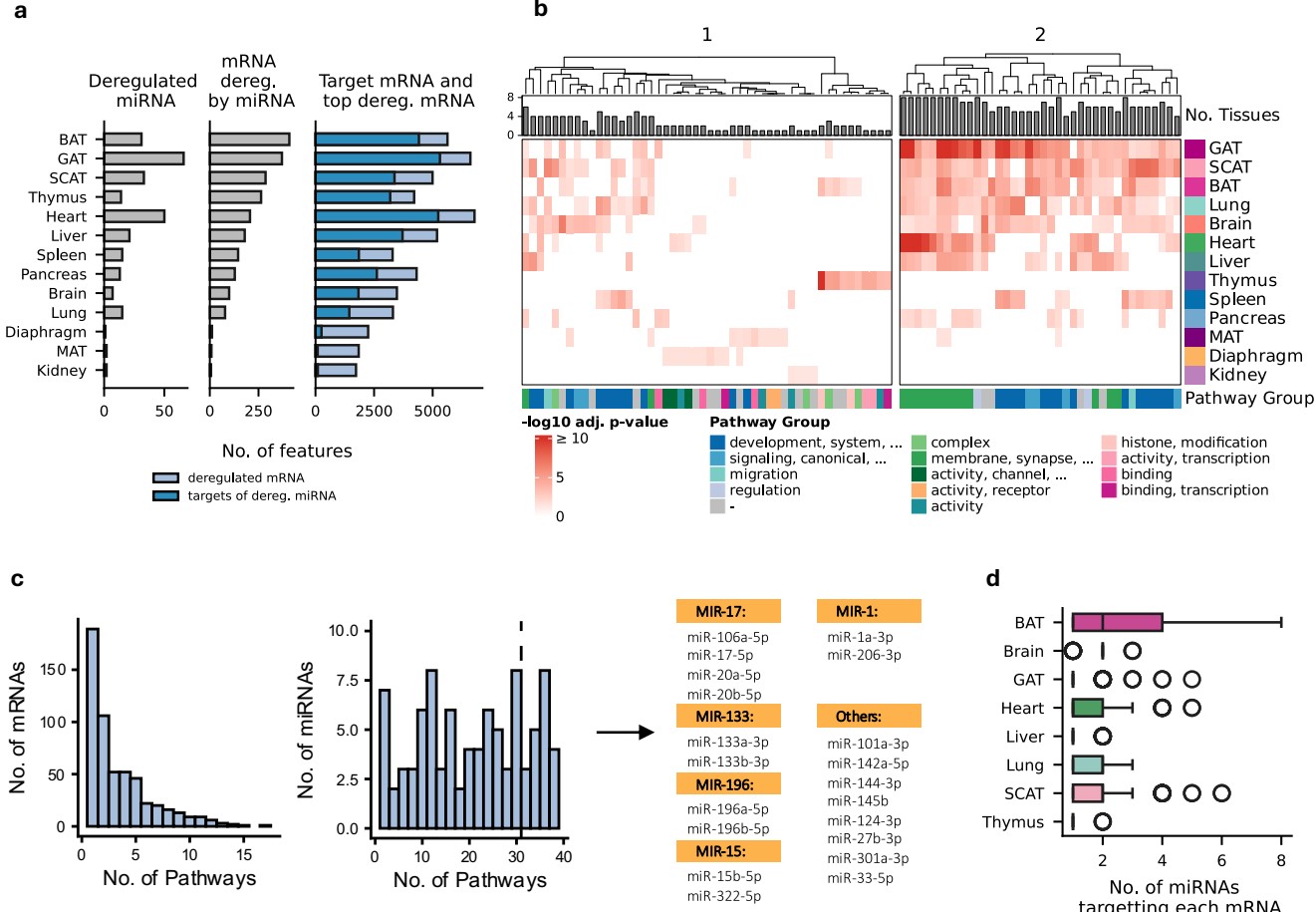

**Fig. 4 | Multi-modal systemic and tissue-specific changes from integrating miRNA and mRNA profiles from scRNA-seq spaceflight experiment.**
**a** Comparison of the number of deregulated miRNA and the top deregulated (see Methods) mRNA, as well as the mRNAs that are targets of the deregulated miRNA and the overlap between those targeted mRNAs and the deregulated mRNAs (mRNA deregulated by miRNA). Source data are provided as a Source Data file.
**b** Top five most-enriched pathways for each tissue (ORA; clusterProfiler; GO: Biological Process; one-sided, BH adjusted) using the top deregulated mRNA that are targeted by deregulated miRNAs as input. Pathways are clustered hierarchically, resulting in sets of tissue-specific pathways (1) and systemic pathways (2). Barplots show the number of tissues in which a pathway was found to be deregulated. Pathways were clustered by similarity in GO, and pathway groups are reported as the most frequent words in the clusters (See method). **c** Distributions of the number of mRNAs occurring per pathway and the number of miRNAs targeting each pathway, respectively, for the systemic pathways from (**b**). The miRNAs that are contributing to at least 30 pathways are shown on the right. MiRNAs are labeled by the MIR family if there are at least two miRNAs from a particular MIR family. Otherwise, they are grouped under "Others". Source data are provided as a Source Data file. **d** Boxplot of the number of deregulated miRNAs that target each mRNA. (Number of replicates see Source Data). Boxplots show the median (center line), the first and third quartiles (box), and whiskers extending to the most extreme data points within 1.5×IQR; values beyond this range are plotted as outliers. Source data are provided as a Source Data file.

cause a negative correlation, due to indirect regulations, co-regulation with other miRNAs, Transcription factors or other factors[25]. Our analysis identified BAT as the tissue with the highest number of deregulated mRNA targeted by deregulated miRNA, totaling 408 mRNAs (Fig. 4a). In total, the miRNA-mRNA pairs miR-132-3p - *RORB*, miR-206-3p - *SMAD9* and miR-378a-3p - *TFCP2L1* were deregulated across most tissues (four each). Notably, all three targets are transcription factors, propagating the miRNA regulation signal in multiple tissues. Interestingly, we found that mRNA deregulation does not necessarily coincide with miRNA deregulation. For example, we observed that the diaphragm, MAT and kidney showed 2001, 1757, and 1635 deregulated mRNAs, but only 1, 2, and 2 deregulated miRNAs. This results in a similarly low number of miRNA-mRNA pairs. These findings suggest that additional regulatory factors, beyond miRNA deregulation, contribute to spaceflight-induced effects. To further explore the functional impact of these changes, we performed a pathway analysis (ORA with clusterProfiler) using the targeted and deregulated mRNAs. From an initial set of 3455 pathways, we extracted the top 10 most enriched pathways per tissue (Fig. 4b). Hierarchical clustering grouped the

pathways into two categories: pathways that affect a low number of tissues (cluster 1) and pathways that were systemically enriched across multiple tissues (cluster 2). Systemic pathway disruption primarily affected SCAT, GAT, BAT, liver, lung, heart, brain and the spleen. The most prominent pathways in this cluster involved development- and the cell membrane-related processes, including "lung development", "epithelium migration" and "presynaptic membrane" (Supplementary Fig. 5a). Thymus, MAT and diaphragm exhibited distinct patterns of tissue-specific pathway deregulation. Pathways deregulated in MAT, diaphragm and kidney affect membrane function, including "cadherin binding" and "adherens junction". The diaphragm displayed disruption in multiple ion channel pathways, including "ion channel inhibitor activity" and "potassium ion leak channel activity". Thymus, however, presented disruption in pathways related to the nuclear environment, such as "regulation of histone modification" and "heterochromatin organization". Of note, biological pathways consist of subcomponents, so while pathways like "bone development" and "lung development" might appear similar, they contain both shared and unique elements.

**a**

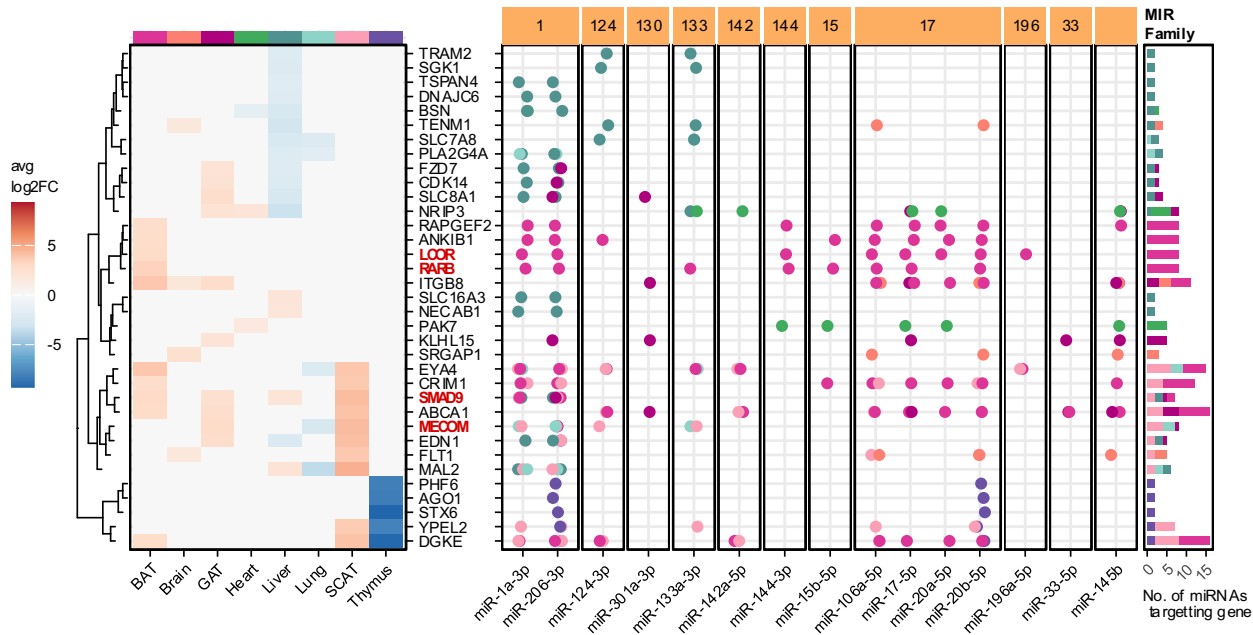

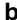

**b**

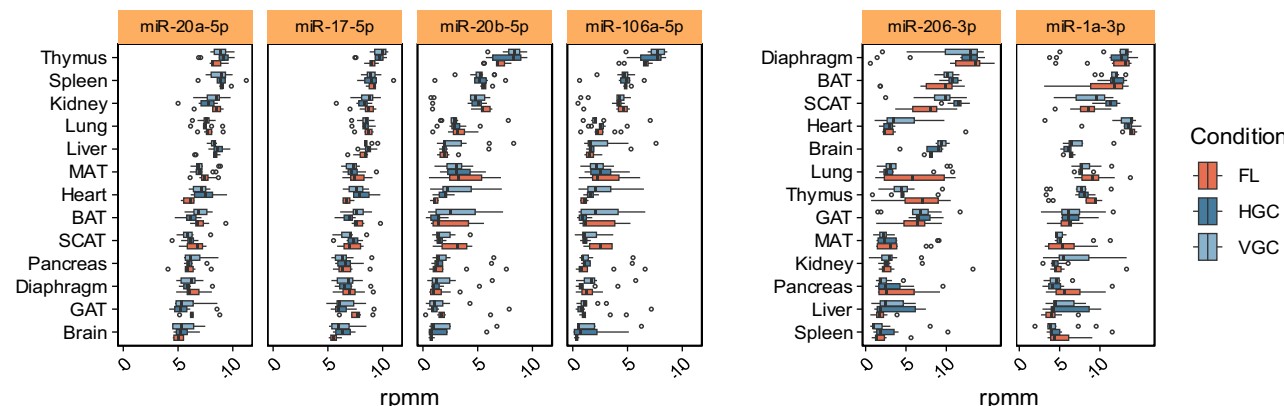

**Fig. 5 | miRNA–mRNA targeting networks and expression changes across tissues. a** Heatmap of average log2 FC between Flight (FL) and Habitat Ground Control (HGC) for each tissue for mRNA most heavily targeted by miRNA (left), along with the miRNAs targeting the mRNA (right). A dot represents targeting, while no dot means no targeting. The color of the dot represents the tissue where the targeting is happening. The miRNAs are grouped based on which MIR family they belong to. Stacked barplots on the right show the count of the number of miRNAs targeting each mRNA in each tissue. **b** Boxplots rpmm expression of miRNAs from MIR-17 and MIR-1 cluster for Flight (FL), Habitat Ground Control (HGC) and Vivarium Ground Control (VGC) with tissues ordered by the median expression in each tissue (Number of biological replicates *n*: See Fig. 1c, Supplementary Data 1) Boxplots show the median (center line), the first and third quartiles (box), and whiskers extending to the most extreme data points within 1.5×IQR; values beyond this range are plotted as outliers. Source data are provided as a Source Data file.

To assess whether genes driving pathway enrichment were shared across similar pathways, we analyzed the distribution of mRNAs enriched in pathways across multiple tissues from cluster 2 (Fig. 4c). Although selected pathways had similar functions, the participating mRNAs were largely distinct, with most appearing in fewer than five pathways. However, four miRNAs regulated mRNAs across all pathways (miR-124-3p, miR-142a-5p, miR-1a-3p and miR-206-3p). We therefore focused on the top 20 miRNAs involved in most pathways. Similar to the systemic pathways, multiple miRNAs such as miR-20b-5p and miR-132-3p were involved in up to 18 different tissue-specific pathways from cluster 1 (Supplementary Fig. 5b). The *MIR-17* family of miRNA was especially prominent, being involved in structural changes and nuclear reorganization by histone modification in BAT and thymus.

The overall similarity between the pathways prompted us to identify target genes of miRNAs contributing to tissue-specific and broadly dysregulated pathways from clusters 1 and 2. Specifically, we identified mRNAs targeted by the largest sets of miRNAs (Fig. 4d). In SCAT, BAT and GAT, the most frequently targeted genes included *ITGB8, FLT1, RARB, MECOM* and *EYA4*, which play key roles in focal adhesion, vasculature and tissue morphology. Those genes are, among others, targets of miRNAs belonging to the *MIR-1, MIR-133* and *MIR-17* families. Notably, four of the 35 most frequently targeted mRNAs encode transcription factors (*LCOR, RARB, SMAD9* and *MECOM*), likely amplifying regulatory signals to other genes (Fig. 5a). Deregulation of miRNAs resulted in mRNA downregulation in the thymus, liver and lung and mRNA upregulation in BAT, SCAT, GAT and brain. To compare miRNA expressions across tissues and conditions, we visualized

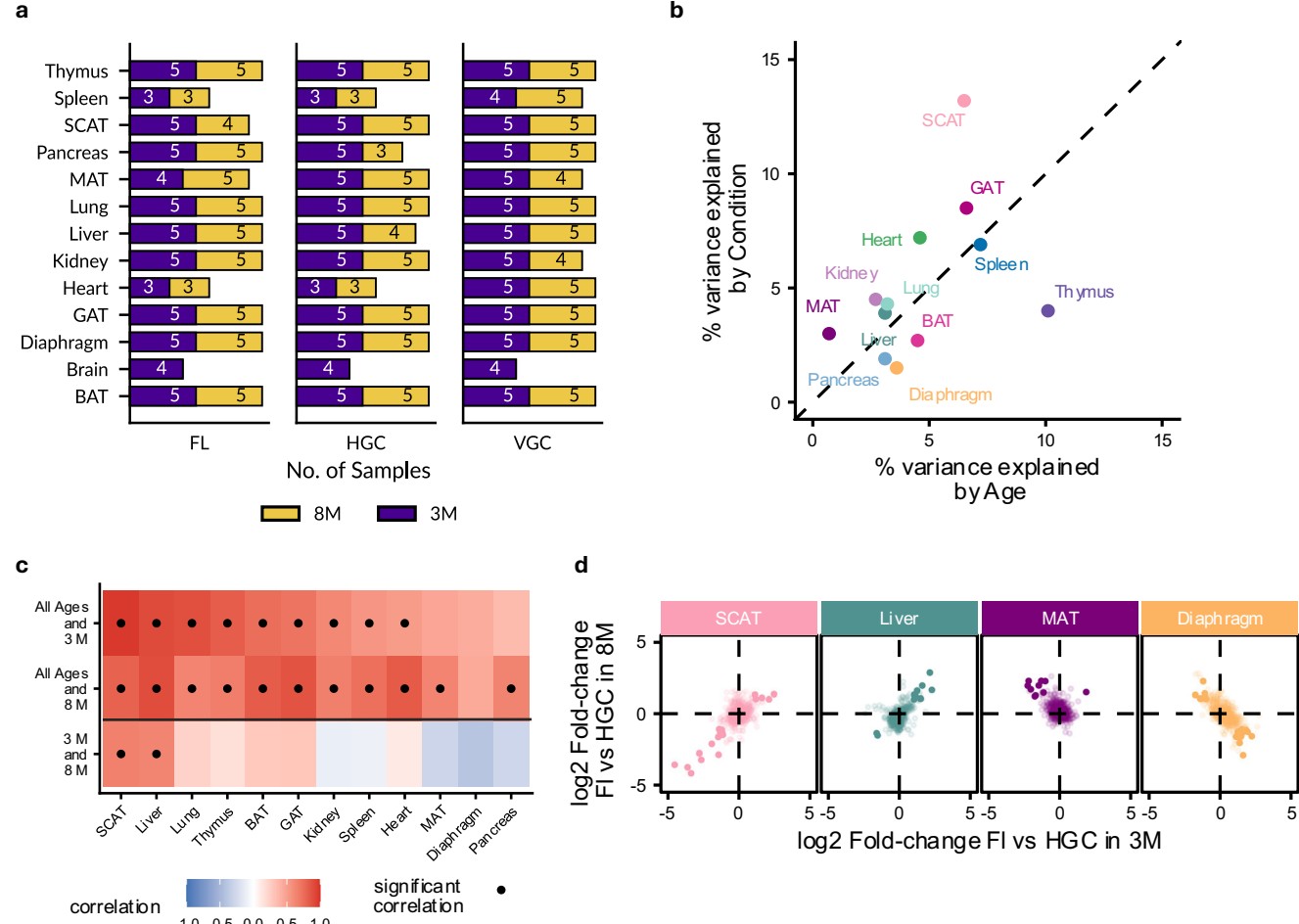

**Fig. 6 | Age-group specific spaceflight effects. a** Sample counts for each age group (3 months and 8 months) for each tissue split by condition, flight (FL), habitat ground control (HGC) and vivarium ground control (VGC) for the live animal return (LAR) group only. **b** Comparison of variance explained by condition and age for tissues from the group. Note, the brain was not included as we only have samples from the 3-month group. Source data are provided as a Source Data file. **c** Correlation between miRNA fold-changes of FL vs. HGC across the age groups 3 months, 8 months, and all ages. Significant correlations (adj. *P* value < 0.05, two-sided test, abs. Pearson's correlation >0.5, BH adjustment) are shown with a dot. Source data are provided as a Source Data file. **d** Effects of spaceflight on the different age groups. SCAT and Liver correspond to examples of tissues with the highest correlation between miRNA fold-changes (Pearson's *R* value = 0.65 and 0.62, see Fig. 6c), while MAT and diaphragm correspond to tissues with the lowest correlation (Pearson's *R* value of = −0.29 and −0.40, see Fig. 6c). Source data are provided as a Source Data file.

members of the *MIR-17* and *MIR-1* families (Fig. 5b). *MIR-17* family members were most abundant in the thymus, but their target genes were primarily dysregulated in the adipose tissues SCAT, BAT, and GAT. However, miR-20b-5p targets the downregulated mRNAs *PHF6*, *AGO1*, *STX6*, *YPEL2*, and *DGKE* in thymus. Members of the MIR-1 family were most abundant in the diaphragm but also targeted the adipose tissues SCAT, GAT and BAT. Notably, miR-206-3p also targeted the same downregulated cluster of genes in the thymus as miR-20b-5p.

Our results so far indicate that spaceflight disrupts tissue-specific miRNA expression patterns, which appear to be correlated with gene dysregulation. Notably, *MIR-17* family members, highly expressed in the thymus, extensively target adipose tissue mRNAs, while *MIR-1* family members, predominant in the diaphragm, also regulate genes in SCAT, GAT, and BAT. In addition, miR-20b-5p and miR-206-3p jointly target a cluster of downregulated genes in the thymus, suggesting coordinated regulatory effects. Since the miRNA-mRNA targeting can be spurious, we further compared our deregulated miRNA-mRNA pairing against Tarbase 9[26] and miRTarbase 2025[27] to see if they have been validated in other studies. Out of 62,754 targets we highlighted, 9569 have previously been experimentally validated in the literature (Supplementary Data 6). Furthermore, a sub-analysis using just the

validated miRNA-mRNA pairs confirmed that our main results remain consistent (Supplementary Fig. 6). As miRNA expression changes with age and spaceflight effects resemble aging, this raises a critical question: do these spaceflight-induced changes differ between young (3-month-old) and middle-aged (8-month-old) mice?

## Spaceflight causes both age-dependent and age-independent miRNA-mRNA regulation

Long-term spaceflight induces degenerative effects on multiple organ systems, assumed to resemble accelerated aging. To determine whether spaceflight indeed affects age-related differences, we analyzed 3- and 8-month-old mice in 12 out of 13 tissues (Fig. 6a). We excluded the brain due to a lack of samples. Based on the percentage of variance explained by age versus the condition using PVCA, the thymus (age: 10.1%, condition: 4%) presented the strongest age effect (Fig. 6b). When analyzing fold-change correlations for FL vs. HGC, the 3-month and 8-month groups yielded significant positive correlations to the overall group including both ages (avg. Pearson's correlation = 0.66, sd = 0.20 and avg. correlation = 0.71, sd = 0.13; Fig. 6c). However, the correlation weakened when comparing 3-month and 8-month directly (avg. correlation = 0.09, sd = 0.34), indicating a shared spaceflight

response alongside tissue-specific and age-dependent effects. SCAT (correlation = 0.65, $p$ value = $1.72*10^{-139}$) and liver (correlation = 0.62, $p$ value = $1.53*10^{-126}$) exhibited a high correlation, indicating a low age-dependence, whereas MAT (correlation = −0.29, $p$ value = $5.25*10^{-23}$) and diaphragm (correlation = −0.40, $p$ value = $1.59*10^{-45}$) showed a weak correlation (Fig. 6d).

Comparison with the TERM group helps distinguish whether these differences stem from reentry-induced stress or prolonged spaceflight exposure. Fold-change correlation between FL vs. HGC in both age groups jointly and in 3-month and 8-month matched between the LAR and TERM cohorts (Supplementary Fig. 7a). However, the specific tissues showing the strongest positive and negative correlations differed between LAR and TERM. Unlike LAR, where MAT, diaphragm and pancreas showed the strongest effects, TERM exhibited the lowest correlation in diaphragm (correlation = −0.21, $p$ value = $1.89*10^{-12}$), BAT (correlation = −0.37, $p$ value = $5.85*10^{-38}$) and GAT (correlation = −0.40, $p$ value = $1.14*10^{-43}$). In this consideration, we excluded the pancreas from TERM due to insufficient sample availability. Taken together, our findings indicate that the age-dependent effects emerge even in the early stages of spaceflight.

Given the observed effects of spaceflight on the age groups, we investigated whether these differences result from accelerated aging. We thus compared our results to age-related miRNA expression patterns in the TMS noncoding RNA dataset[18]. This approach enabled a comparison of the age-related changes in spaceflight-exposed mice to those in mice physiologically aged up to 21 months (Supplementary Fig. 7b). Because our cohort includes female mice, we filtered the TMS accordingly to match sex and tissue type (Supplementary Fig. 7c). Notably, the *M. musculus* strain for our spaceflight study (BALB/cAnNTac) differed from TMS (vC57BL/6JN). As miRNA expression is not only tissue and age-specific, but also strain-specific, this fact adds noise to our comparison. We used 3-month-old mice as the baseline for TMS, compared it against progressively advancing age and considered the overlap with our 8-month vs. 3-month comparisons in HGC-control and Flight (Supplementary Fig. 7d, Supplementary Data 7). In spaceflight-exposed mice, the overlapping miRNA sets expanded with age, reaching their maximal size at 21 months in the GAT and pancreas. The TMS comparison 9-month vs. 3-month is closest to our 8-month vs. 3-month control comparison, which is why we expect larger overlaps for this time-point. However, a peak around the 9-month time point for the Control group was absent. To identify overlaps with the aging-associated miRNAs, we compared the miRNAs deregulated in at least three time-points in TMS with those altered in 3-month vs. 8-month across FL, HGC and VGC (Supplementary Fig. 7e). This analysis resulted in three miRNAs (miR-92a-3p, miR-322-5p and miR-205-5p) yielding the same changes in TMS and the control groups. In the Flight-group miR-92a-3p followed an age-consistent trajectory, while miR-322-5p and miR-205-5p displayed expression patterns distinct from physiological aging. Comparison with the TMS cohort revealed complex patterns in age-dependent differences between spaceflight and control groups, highlighting the need for further investigation using mice from the same strain and a larger age span.

To gain deeper insights into the biological mechanisms underlying age-specific miRNA changes in space from the LAR group, we integrated the mRNA data. We identified mRNAs that both targeted deregulated miRNAs and exhibited differential expression between 3-month-old and 8-month-old mice in Flight and HGC to clarify spaceflight-induced aging effects (Supplementary Fig. 8a). Pathway analysis (GSEA with miEAA) revealed that most top-enriched pathways were shared between FL and HGC (Supplementary Fig. 8b). These findings indicate that age-related differences in Flight and HGC were similar, suggesting they do not account for the observed negative correlations between FL vs. HGC in 3-month and 8-month. We thus removed effects of spaceflight that influenced 3-month and 8-month similarly, isolating the age-dependent spaceflight effects. We defined

two miRNA groups: named Flight 3-month and Flight 8-month. We classified Flight 3-month miRNAs as those deregulated in both FL vs. HGC in 3-month and in 3-month vs. 8-month in FL, consistently showing upregulation or downregulation in FL 3-months. We defined Flight 8-months miRNAs analogously using the FL vs. HGC in 8-months and in 3-months vs. 8-months in FL comparisons (Supplementary Fig. 8c). Pancreas, diaphragm and MAT, which previously had the lowest correlations presented with the highest number of age-dependent miRNAs under spaceflight (pancreas: 116, diaphragm: 94 and MAT: 37) (Fig. 7a). We performed the same analysis for the top deregulated mRNAs and computed the overlap with their miRNA targets, using those as input for a pathway analysis (ORA with clusterProfiler). While clustering of the top three pathways per tissue (Fig. 7b) yielded few membrane-related pathways shared across multiple tissues (MAT, GAT, and diaphragm), the majority of the age-dependent spaceflight pathways remained tissue-specific. For example, the kidney was enriched for "ECM structural constituent conferring tensile strength," and MAT showed enrichment of "regulation of cell morphogenesis involved in differentiation" in 3-month-old. Since miRNAs are often organized into families with shared sequence and functional similarities, we next examined which MIR families contributed most to age-dependent spaceflight effects.

We identified MIR families most involved in age-dependent spaceflight effects by selecting those with at least two deregulated members in a tissue. Here, *MIR-8*, *MIR-154* and *MIR-15* stand out as most represented, predominantly associated with age and spaceflight in the diaphragm, pancreas and MAT (Fig. 7c, d). Notably, most MIR family members in each tissue display consistent deregulation in the same direction under the same condition. As we did not explicitly select for this effect, those results suggest a highly coordinated regulation. Among 49 mRNAs targeted by age-dependent spaceflight-affected miRNAs, we identified nine transcription factors, such as *TEAD1*, *RORB*, *MYB*, and *MEIS1* (Fig. 7e). As the heatmap shows, some of the mRNAs follow age-dependent patterns. We observed that most miRNA targeting of these mRNAs originated from the MAT, diaphragm and pancreas, particularly in 3-month-old mice rather than in 8-month-old. For instance, members of *MIR-15* and *MIR-17* families targeted transcription factors *RORB* and *MYB* in the diaphragm in 3-months but not in 8-months (Supplementary Fig. 8d). This suggests age-dependent regulation by miRNA from multiple targets as a result of spaceflight. We also compare the miRNA-mRNA targeting we highlight here against Tarbase 9 and miRTarBase 2025, confirming that our conclusion remains consistent even when using only experimentally validated targeting (Supplementary Fig. 9, Supplementary Data 8).

Taken together, our findings reveal that certain miRNAs, such as those from the *MIR-8*, *MIR-15* and *MIR-154* families, show age-dependent changes in diaphragm, MAT and pancreas in response to spaceflight. However, overall miRNA deregulation patterns suggest a coordinated age-independent response to spaceflight stress. Despite some overlap with natural aging signatures, spaceflight-induced changes follow distinct regulatory trajectories, highlighting the need for further studies to disentangle adaptive versus degenerative processes in different tissues. These findings are crucial in the context of extended space missions, as understanding age-dependent molecular responses to spaceflight will be essential for interplanetary travel. Our results do not indicate a strong acceleration of aging in spaceflight-exposed mice. If this also holds true for humans, even older adults may remain viable candidates for long-duration missions, such as a journey to Mars.

## Discussion

To study the effects of spaceflight on *M. Musculus*, we sequenced 686 samples of 13 organs from mice sent to the ISS and ground controls. This study examines how spaceflight impacts different murine

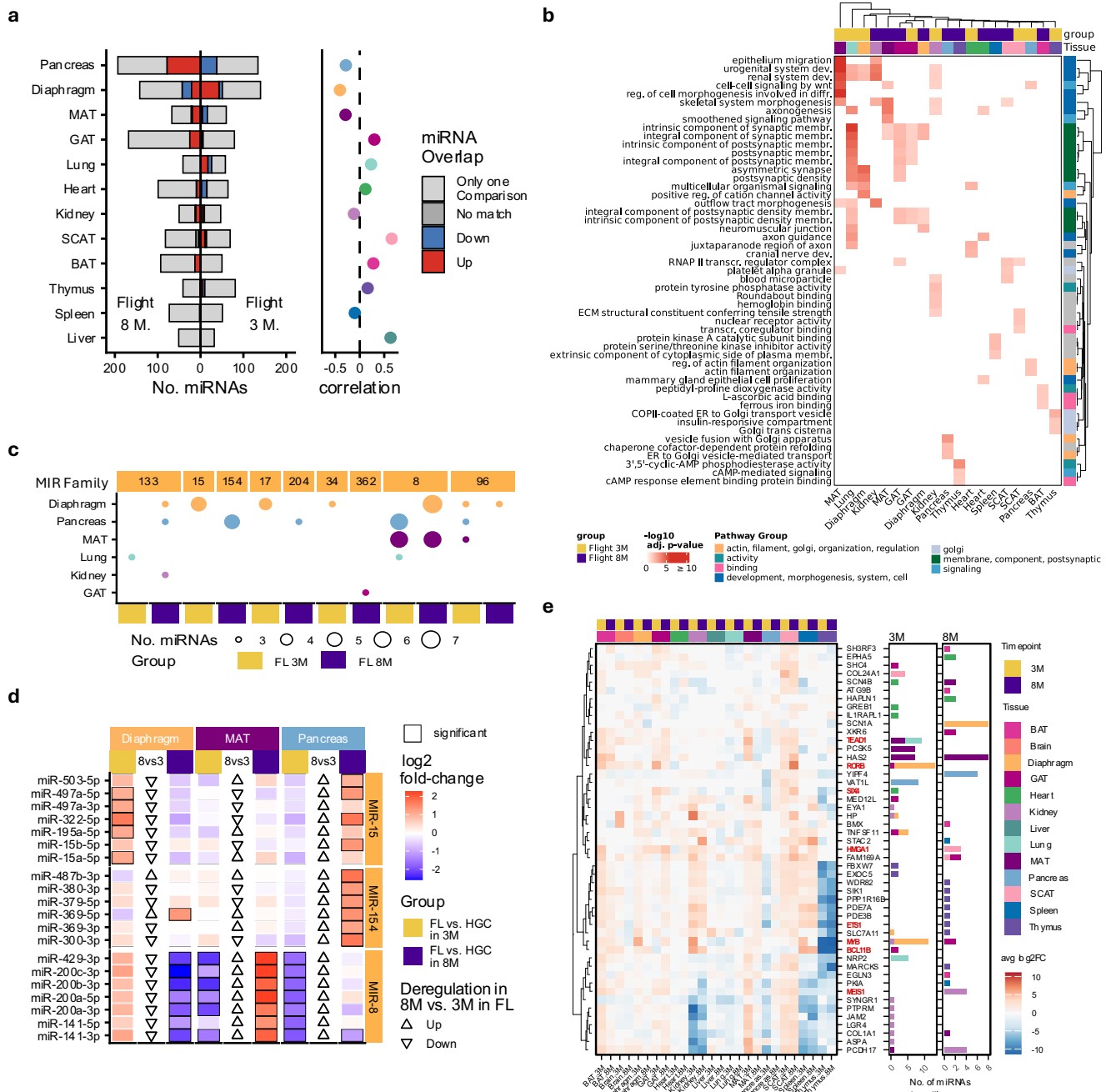

**Fig. 7 | Age- and condition-specific miRNA–mRNA interactions and pathway impacts. a** Number of deregulated miRNA with equal direction (up or down), not equal direction (no match) or only occur in one comparison in the age or the condition comparison (only one comparison). Flight 3 month denotes the miRNAs that are occurring in 3 M vs. 8 M in flight, and in FL vs. HGC in 3 M, and flight 8 month denotes the miRNAs that are occurring in 8 M vs. 3 M in flight, and in FL vs. HGC in 8 M. The correlations were derived from Fig. 6c. Source data are provided as a Source Data file. **b** Pathway analysis (ORA, clusterProfiler; one-sided, BH adjusted) of the mRNAs that are targeted by the miRNAs from (**a**) and that fulfill the same restrictions as the miRNA in (**a**). Pathways were clustered using the similarity in GO, and the pathway groups were labeled with the most frequent words (See method.)

**c** Number of miRNAs from (**a**) for each tissue grouped by the MIR Family of origin, for all MIR families with more than two miRNAs in one tissue. Dot color corresponds to the tissue color, while FL 3 M and FL 8 M groups are in yellow and violet. Source data are provided as a Source Data file. **d** Heatmap of log₂ fold-change in FL vs. HGC in 3-months and in 8-months of MiRNAs from the three most frequent MIR-families from (**a**). The triangles show the direction of deregulation in 3 months (3 M) vs. 8 M in flight. **e** Heatmap of log₂ FC for FL vs. HGC in 3- and 8-month-old mice for the mRNA set that is the most heavily targeted by miRNA. Transcription factors are indicated in bold red text. The stacked barplots on the right indicate the number of miRNAs targeting a particular mRNA in the corresponding tissue in 3-month flight or an 8-month flight. Source data are provided as a Source Data file.

tissues and extends current knowledge on molecular spaceflight impact. Previously, Beheshti et al.[15] analyzed RNA-seq data from the NASA Genelab database. Their study of liver, kidney, adrenal gland, thymus, mammary gland, skin and multiple skeletal muscles identified miRNA signatures interacting through the *TGFB1* and *p53* genes to influence adaptation to microgravity and space radiation. This miRNA

signature was further explored in the circulating miRnome[16]. We built upon these studies by sampling a wider variety of organs from the same mice while also considering housing effects on the miRNA/gene expression. Although double-density housing minimally affects mice[28], ISS conditions with elevated $CO_2$ concentration, different temperature, and humidity, impact their physiology and behavior[29].

In our data, housing effects had the most profound impact on brain, kidney and MAT, likely due to $CO_2$ and temperature influences on brain and kidney function[30,31]. After accounting for the housing effects, our data suggested that miRNAs primarily regulate tissue remodeling and DNA damage repair pathways at a systemic level. Tissue remodeling likely resulted from the sensing of mechanical stressors such as gravity (or lack of), which stimulates ECM modifications and propagates signals to differentiating cells or developmental pathways[32]. This effect was most prominent in GAT, SCAT, pancreas and spleen, suggesting adipose tissue remodeling driven by metabolic changes resulting from the peripheral insulin sensitivity[33] seen in spaceflight[34]. Furthermore, we found miRNAs associated with adipocyte function, such as miR-200b/a[35] and miR-196a[36], in pancreas, GAT and SCAT, with miR-196a also contributing to ECM reorganization. Given that pancreatic hormones regulate adipocytes, and spaceflight increases the mass of pancreatic beta-cells[37], this is further evidence of metabolic dysfunction. On the other hand, DNA damage repair pathways emerged as DNA metabolism and replication in multiple tissues. Specific miRNAs involved included members of the *MIR-1* family, miR-1a-3p and miR-206-3p, a family reported to play key roles in cancer and as a tumor-suppressor[38]. Spaceflight profoundly affects the cardiovascular system[21], with miRNAs in our dataset appearing to regulate heart innervation through changes in the postsynaptic membrane and potassium. These findings suggest a potential role for miRNAs in the QT interval prolongation or arrhythmias observed in astronauts. MiRNAs previously linked to coordination of synaptic cells include members of the *MIR-34* family, such as miR-34c-5p, miR-34b-5p[39] and miRNAs from *miR-1/133*[40], such as miR-133a-3p, miR-133b-3p and miR-1a-3p.

The scRNA-seq data also suggest that ECM disruption is a key feature of cell-type-specific response to spaceflight. Since two independent modalities yielded similar results, we confidently integrated the datasets to explore miRNA-mRNA regulation in detail. This allowed us to categorize the pathways into systemic and more tissue-specific effects. For the systemic effects, we primarily observed tissue remodeling through ECM developmental pathways, synaptic membrane modification. ECM remodeling and developmental pathways were extensive in the GAT, SCAT, BAT and liver. One prominent example of miRNAs involved in the systemic changes is the *MIR-17* family, previously shown to play a role in adipocyte differentiation[41], in particular through the Wnt signaling pathway[42]. Interestingly, we also saw the Wnt signaling pathway deregulated in GAT, SCAT and BAT, suggesting the involvement of this miRNA family. It is also known to influence adipose differentiation through the ECM, which we saw deregulated in multiple tissues across miRNA and mRNA data[43].

We also observed synaptic membranes deregulated in the Brain, along with the ECM dysfunction. Members of the *MIR-17* family, such as miR-20b-5p, played a central role. Notably, miR-20b is involved in Alzheimer related pathways in humans[44], raising the possibility that spaceflight-induced metabolic dysfunction contributes to cognitive impairment[45]. For DNA damage, we saw strong signals of nuclear environment remodeling, likely as a response to radiation exposure. Members of the *miR-17/92* cluster, such as miR-92b and miR-363-3p[46] are involved in radiation exposure-related DNA damage control. Additionally, LET-7, which influences both systemic and more tissue-specific pathways, plays a well-documented role in radiation response, including exposure to deep-space radiation[16,47]. A small set of miRNAs, such as from the *MIR-17/92*, *MIR-34* and *MIR-1/133* families, seems responsible for much of the dysfunction that we see. These miRNAs impacted multiple pathways, yet their targets were not frequently shared between pathways or tissues. However, among the most heavily targeted mRNAs, tissue remodeling and DNA damage showed up as recurring themes. For BAT, targets included *ITGB8*[48] and *CRIM1*[49] known to be involved in ECM or membrane modification. Genes such as *BSN*[50] and *TSPAN4*[51] were targeted in the liver, which are also known

to have functions related to the cell membrane. DNA damage was particularly noticeable in the thymus, where *PHF6*, a transcriptional regulator mutated in leukemia[52,53], was a miRNA target. *DGKE*, another mRNA targeted in the thymus, is associated with hemolytic uremia[54]. Interestingly, the limited observation of DNA damage response is somewhat counter to what is seen in circulating miRNAs, where this response dominates. While this might be partially due to the different strains of mice used in the study, it might also hint at differing response times of blood and solid tissue to the ionizing radiation prevalent on the ISS, as solid tissue is known to have slower response kinetics in comparison to blood[55]. However, due to the absence of blood samples, this hypothesis cannot be verified from our data directly. Interestingly, both ECM dysfunction and DNA damage are hallmarks of aging. In fact, spaceflight-induced degeneration is often considered analogous to age-related dysfunction, as organisms become more frail and less resilient to stressors over time.

We finally examined whether the spaceflight effects differ by age. While age-related variance was smaller than that derived from spaceflight, it was still present. Pancreas, Diaphragm and MAT showed up as the most prominent tissues to have an age-dependent component as a result of spaceflight. Tissue remodeling was still present when we took the effect of age into account, although the response to DNA damage was less prominent. Key miRNA families involved in age-dependent spaceflight effects included *MIR-8*, *MIR-15* and *MIR-154*. *MIR-8* miRNAs, found in Diaphragm, MAT and Pancreas, regulate neural cell development and homeostasis[56,57]. The *MIR-15* miRNAs in the diaphragm indicate muscle atrophy[58], especially in the 3-month-old mice. Both *MIR-15* and *MIR-154*, detected in pancreas, have been linked to cancer[59–61] indicating a change in tissue homeostasis in the 8-month mice. When considering age, endocrine and immune-related function became more pronounced. For instance, we found age-specific targeting of *YIPF4* in the pancreas, a gene involved in nutrient stress-related macroautophagy[62], suggesting potential differences in the post-translational processing of the primary endocrine hormones like insulin. We also saw an age-related change in the diaphragm. In 3-month mice, we saw developmental pathway deregulation, whereas in 8-month mice, deregulation primarily affected the synaptic membrane. Notably, miRNAs in the 3-month diaphragm targeted *MYB* and *RORB*, transcription factors essential for stem cell and neuron development[63,64], but these interactions were absent in the 8-month-old. Similarly, *TNFSF11* (*RANKL*), targeted by miRNAs in the *MIR-17* family such as miR-20a-5p and miR-17-5p in 3-month and linked to skeletal muscle development and function, was not targeted in 8-month[65,66]. This suggests that developmental pathways become less accessible for adaptation in the older mice. Additionally, miR-15, which regulates them both, interacts with *MYB* in natural killer cell maturation, a type of cell involved in the regeneration of skeletal muscle[63,67]. Similar immune system-related mRNAs were similarly implicated in adaptation for the 3-month but not 8-month mice, reinforcing the idea that younger mice maintain greater adaptability to spaceflight-induced stressors.

While our findings align with existing literature, we acknowledge the broader challenges associated with spaceflight research, which extend beyond our study. Conducting experiments in space involves inherent logistical constraints, particularly in housing and retrieving mice, which naturally limit sample sizes. Taking the brain as a whole during library preparation is likely to remove brain region-specific effects of spaceflight, as several miRNA are known to show localized expression[68]. While our highlighted miRNA mmu-miR-20b-5p was not found to be brain-region specifically expressed, it is worth considering the known effect of strain on miRNA expression, as our previously published mouse brain aging atlas[68] contains C57BL/6JN vs. BALB/cAnNTac used in this study. It might be that the deregulation observed here is resulting from the combined deregulation in multiple regions or in only a single region. Similarly, circulating miRNAs could provide a

systemic view of the tissue-specific changes observed. Although circulating miRNAs were not collected in this study, comparison to published spaceflight datasets and reference expression profiles indicates that many of the tissue miRNAs affected here (49 out of 56) are detectable in blood, plasma, or serum, suggesting potential systemic biomarker relevance (Supplementary Data 9). Despite these challenges, we took extensive measures to optimize our study design, prioritizing robust changes with the largest effect sizes that remained consistent across different experimental conditions and modalities, rather than relying solely on conventional adjusted $p$ values[69,70]. Additionally, our cohort consisted exclusively of female mice, which reflects common logistical considerations but also presents an opportunity for future research to explore sex-specific effects, an area of increasing importance in the scientific community. Furthermore, while the age range of our mice corresponds to young adult to middle-aged humans, aligning well with the typical astronaut demographic, aging biomarkers tend to emerge later in a mouse's lifespan. Although this prevented us from assessing certain late-stage aging markers in response to spaceflight stressors, our study still provides valuable insights into physiological adaptations relevant to space missions. By carefully addressing these challenges, we contribute to the broader effort of refining spaceflight research methodologies and expanding our understanding of biological responses to space environments.

Overall, our study uncovered the influence of environment and age on miRNA signals related to spaceflight. By integrating single-cell mRNA data, we uncovered the regulatory role of miRNAs in spaceflight adaptation. We have attempted to partially validate the miRNA-mRNA interactions by comparison against experimentally validated data in publicly available websites to see if our conclusions change. Due to the incomplete nature of miRNA-mRNA experimental validation data and the existence of false positives in predicted targets, we were able to verify a fraction of the targeting. Interestingly, our final conclusion remains unchanged. Validating these findings in a true spaceflight environment more extensively remains challenging. However, since most observed effects involved ECM and structural components, further studies could use cell lines, such as endothelial cells in simulated microgravity, to check if controlling the expression of *MIR-17/92*, *MIR-1/133*, *MIR-15*, or *MIR-8* families mitigates spaceflight-induced structural changes. This dataset serves as a resource to further study organ changes in mice in space and on Earth, providing a foundation for more detailed experiments.

## Methods

### Samples and cohorts

This experiment was approved by the NASA Flight Animal Care and Use Committee (Protocol #FLT-18-116) at the NASA Ames Research Center (Moffett Field, CA) and the Kennedy Space Center (KSC, FL). The Flight mice were received after landing at Scripps Research Institute (09-0004, for animal receipt upon landing) and transported to Stanford University IACUC (WYS1591).

For the Rodent Research Reference Mission 1 (RRR-1)/Rodent Research-8 (RR-8) 40 BALB/cAnNTac female mice (Taconic Biosciences) were sent to the ISS (ISS) aboard SpaceX-16 on 12/5/2018 (Flight mice). As control groups, two groups of 40 mice each were kept on Earth. These mice were assigned to VGC and HGC. In each group, ~3-month-old and ~8-month-old mice were used, representing different age groups. Eight of those mice (two each from LAR 3 M, LAR 8 M, VGC 3 M, VGC 8 M) were processed and used for the scRNA-seq data (Supplementary Information).

VGC mice were housed at NASA Kennedy Space Center in standard vivarium conditions at 4 mice per cage, 20–22.2 °C, 12 hr light 12 hr dark. HGC mice housing was matched to the conditions experienced by Flight mice (as closely as possible) by using double-density housing (the same Rodent Habitats (AEM-X) and Animal Enclosure Module Transporters (AEM-T) as the flight animals), with identical

food, matched temperature, humidity, and carbon dioxide levels based on actual flight environmental monitoring. Each habitat contained ten mice (five per side) and included enrichment huts as part of the AEM-X configuration. To dissect all mice together in matching conditions, VGC and HGC mice were transported to Scripps Institute just before the return of the Flight mice and were dissected in matching conditions (+/− 1 day).

From each of the main experimental groups, half of the mice were assigned to the ISS Terminal (ISS Term) subgroup, which were sacrificed onboard the ISS (Flight) or on Earth (VGC, HGC) 22–24 days after launch. The other half were assigned to the Live Animal Return (LAR) group, which was returned alive to Earth on 1/14/2019 after 39.5 days on the ISS, and collected 2-4 days after splashdown, following functional assays at Scripps Research Institute.

All mice were fed with NASA Nutrient Food Bars, starting four weeks before launch. The bars were exchanged weekly. Mice had ad libitum access to both food and water.

During the mission, we encountered the following issues, altering the planned experimental design, which must be considered upon interpretation of results.

1) Within 24 h of the planned launch, mold was found on the food bars, affecting both Flight and HGC mice to an unknown degree. It is possible that the mice were able to eat around the mold. The late replacement of the food bars, now secured with double-sided tape instead of epoxy, caused the bars to free-float during launch and return. The bars of the HGC mice fell to the bottom of the cage. Although the Flight mice appeared healthy upon arrival at the ISS, this might have induced additional stress. Due to concerns about mold, as more was discovered aboard the ISS, the ISS Term mice were sacrificed after ~3 weeks of microgravity rather than the planned 8 weeks, and ground controls were sacrificed to match this new timeline.

2) Due to poor weather, the return of the Flight LAR mice was delayed, and the mice spent 7 instead of the normal 2–3 days in double-density housing. This housing is more confined (10 mice per cage side instead of the 5 in the normal habitat), video monitoring was not possible, and food bars likely became unsecured and were free-floating. Conditions of the HGC mice were matched to these conditions, likely causing additional stress for both groups. Due to the weather, the Flight group likely experienced rough seas upon splashdown and shipping back to shore, which was not controlled for in the HGC group. All tissue collection and handling procedures were performed under NASA IACUC protocol #FLT-18-116 and Scripps protocol #09-0004, following the Rodent Research Reference Mission 1 (RR-8) operational guidelines. Further details about Rodent Research Reference Mission 1 can be found at (https://osdr.nasa.gov/bio/repo/data/payloads/RRRM-1%20%28RR-8%29).

### Tissue dissociation and sample preparation

LAR mice were anesthetized with ketamine/xylazine and euthanized by means of cardiac puncture blood draw (exsanguination), and double thoracotomy. The liver was removed, weighed, dissected into individual lobes (1- left/right medial, 2- left lateral and 3- remaining lobes) and snap frozen in liquid nitrogen. All tissues were collected and frozen within 10–16 minutes of the thoracotomy procedure. All groups were treated identically. Organ samples were maintained at −80 °C or lower temperatures throughout the storage, and during transportation and shipment.

TERM mice were anesthetized with ketamine/xylazine/acepromazine at a dose of 120/15/3 mg/kg and euthanized by means of cardiac puncture, blood draw (exsanguination), and double thoracotomy. The spleen was dissected fresh from each carcass and placed in a tube containing RNA Later. The carcass was then wrapped in foil and frozen in MELFI on board the ISS or in ground −80 °C freezers. Carcasses were

maintained at −80 °C or lower temperatures throughout the storage on board the ISS, in ground freezers and during transportation and shipment. The mice were brought back to earth, where dissection was held. It should be noted that carcasses frozen whole onboard the ISS were free-floating and thus in direct contact with air. This prevented them from snap freezing like samples collected on Earth, which were placed directly in liquid nitrogen. Carcasses were thawed at room temperature one at a time (staggered) until the tissues were sufficiently pliable for dissection (approximately 60-90 minutes). Once thawed, multiple tissues were collected from each carcass within 30 min of the first tissue collected. The liver was removed from each carcass, weighed, dissected into individual lobes (1- left/right medial, 2- left lateral and 3- remaining lobes) and snap frozen in liquid nitrogen.

### Transportation and shipment
Organ samples were maintained at −80 °C or lower temperatures throughout the storage, during transportation and shipment.

### RNA extraction
The miRNeasy-96 protocol was followed for all samples with some modifications, as follows.

Tissue Preparation: To avoid overloading the miRNeasy-96 columns, some frozen samples were trimmed to smaller sizes. All kidney, liver, and spleen samples were trimmed to ~25 mg due to their high RNA content. Pancreas samples were trimmed to ~25 mg if they exceeded 45 mg. GAT, MAT, and SCAT samples >200 mgs were trimmed to ~100 mgs as samples of large size homogenize poorly in 96-well format, even though adipose tissues have relatively low RNA content. BAT and lung samples >100 mgs were trimmed to ~50 mgs, and thymi >50 mgs were trimmed to ~25 mgs. The diaphragm did not require trimming, and whole hemi-brains and whole hearts were homogenized by hand with the TissueRuptor in 1 ml QIAzol per 50 mgs tissue, since trimming samples in a reproducible way to capture the same organ subregion for each would be difficult.

Bead homogenization: All samples, including those previously homogenized with the TissueRuptor, were randomized in 96-well format. Two 96-well plates were run simultaneously as follows. 96-well collection microtube plates were chilled on dry ice, and trimmed tissue samples were placed at the bottom of each well. One pre-chilled 5 mm stainless steel bead was then placed on top of the tissue samples with a bead dispenser. 700 μl pre-chilled (4 °C) QIAzol was then added to each well, or 700 μl QIAzol homogenate from the pre-homogenized heart and brain samples. Plates were immediately homogenized on the TissueLyser II for a total of four 5 min runs at 25 Hz, and plates were rotated and switched between the two arms of the TissueLyser II between each run. Plates were then left at room temperature for 5 min before pelleting debris at 6000 × g for 1 min at 4 °C.

Phase Separation & RNA collection: Debris- and bead-free homogenate was then transferred to a new plate, to which 140 μl chloroform was added to each well, and plates were shaken vigorously for 3 min at room temperature before being spun at 6000 × g for 4 min at 4 °C. 300 μl aqueous phase was then transferred to a new S-Block, to which 525 μl 100% EtOH was added and mixed by pipetting, followed by a room temperature spin at 6000 × g for 30 s. The entire volume was then transferred to a miRNeasy-96 plate on top of an S-plate, sealed with AirPore tape sheet, and spun at 5600 × g for 4 min at room temperature. Flow through was discarded, and columns washed with 800 μl RWT and spun as above. This was repeated with 800 μl RPE, and a final 800 μl RPE wash was followed by a 10 min spin. Finally, 50 μl RNase-free water was added to each column and incubated at room temperature for 1 min, then spun at 5600 × g for 4 min at room temperature. The elution was repeated with a second volume of 50 μl RNase-free water. RNA samples were stored at −80 °C.

### Library preparation and sequencing
All samples were processed using the MGIEasy Small RNA Library Prep Kit with the high-throughput MGI SP-960 sample prep system according to the manufacturer's protocol to generate sequencing libraries. In short, after 3'- and 5'-adapters were ligated to the RNA, a reverse transcription step followed, in which samples were barcoded using the barcodes 1–4, 13–16 and 25–32. In a 21-cycle PCR, the cDNA was amplified and further purified via size-selection through AMPure Beads XP (Beckman Coulter, Germany). The success of this purification was controlled using the Agilent DNA 1000 Kit (Agilent Technologies), and concentrations were measured with Qubit™ 1X dsDNA High Sensitivity kit (HS) (Thermo Fisher Scientific). Sixteen samples were pooled into one library in an equimolar fashion. All 45 libraries were circularized and sent to Hong Kong for sequencing. Single-end sequencing on the BGISEQ500RS with the High-throughput Sequencing Set (SE50) was performed with all samples and provided by BGI, Hong Kong.

### Pre-processing
All miRNA datasets, for Spaceflight and TMS, were processed in an identical manner. Fastq files containing raw sequencing reads were mapped against GRCm39 and miRBase (version 22.1)[71] in order to profile miRNA counts using miRmaster (version 2.0)[72]. Information about the miRNA family was obtained from mirgenedb 3.0[73]. Detection filtering was performed by keeping only miRNAs with at least five raw reads in at least 10% of samples in each condition-tissue group. Expression data was normalized using rpmm normalization (reads per million mapped).

### Principal variance component analysis (PVCA)
The principal variance component analysis was performed on the $\log_2$-transformed, filtered rpmm values, with proportion of the variation to be covered by top k PCs at 90%. Factors that were considered were the age, tissue, space (Flight, Not Flight) and Environment (VGC, Not VGC) and combinations of them. For the plots showing only 2 factors, only the named factors were considered. The Brain was excluded from the comparison between Condition and Age because it only had one age group in LAR.

### Dimensionality reduction
Two-dimensional embedding was computed using the umap function in the umap-learn package (version 0.3.10) using the $\log_2$-transformed, filtered rpmm-normalized counts as input.

### Processing of the single-cell RNA-seq data (GEO: GSE295428)
Reads were aligned to mm10 (GRCm38) using Cell Ranger (version 5.0.0) using the 10x mm10 reference. Ambient, cell-free mRNA contamination was estimated and corrected with SoupX (version 1.5.2). Samples with an estimated contamination fraction >10% were excluded. For retained samples, adjustCounts() was applied with default parameters. Raw UMI count matrices were imported into Seurat v4.0.3. Cells were removed if they had <200 or >2500 detected features (genes) or >5% mitochondrial RNA content. Samples with <50 cells after QC were excluded. Doublets were identified with DoubletFinder (version 2.0.3). We performed a parameter sweep (paramSweep_v3/summarizeSweep) to select pK per sample; expected doublet rate was set to 5% and pN = 0.25. Cells predicted as doublets were removed. For each sample, counts were log-normalized (Seurat::NormalizeData), 2000 HVGs were selected (Seurat::FindVariableFeatures, "vst"), and data were scaled (Seurat::ScaleData). Samples were integrated with Seurat's anchor-based workflow (FindIntegrationAnchors/IntegrateData) using 30 dimensions (dims = 1:30). PCA was computed on the integrated assay; the number of principal components (30 PCs) was chosen by elbow plot inspection. UMAP

(RunUMAP) provided a 2D embedding. Graph construction and clustering used FindNeighbors and FindClusters with resolution = 0.8. Clusters were annotated by expert manual curation using canonical marker genes; sub-clustering was performed where necessary. Each cluster/cell was mapped to a Cell Ontology class (where available) to facilitate cross-study comparisons. The Skin tissue was removed post-annotation due to a lack of expected marker expression. For indicated comparisons, DE was performed with Seurat::FindMarkers using the Wilcoxon rank-sum test and including sample as a latent variable. Genes were considered significant at Bonferroni-adjusted $p < 0.05$ and $|log2FC| > 0.5$. GSEA was performed using GeneTrail (version 3.2) with the ranked gene list ordered by $-log10$(adjusted $p$) and $|log2FC|$ (descending) using the GO BP/CC/MF, KEGG, Reactome, WikiPathways and Pfam gene sets.

### Differential expression analysis of miRNA and mRNA

Differential expression analysis of the miRNAs was performed using the filtered rpmm values with a Wilcox Rank Sum Test, and $p$ values were adjusted using Benjamini-Hochberg (BH). Fold-changes were calculated using the geometric medians in the two groups. MiRNAs were considered to be deregulated when they showed an absolute $log_2$ fold-change bigger than 1 and an absolute Cohen's $d$ above 0.5.

For the single-cell mRNA pseudobulks, the data were transformed into a Pseudobulk dataset using the aggregateData function (fun = "sum") of the muscat package (version 1.12.0)[74] on the raw count data. The differential expression analysis was then performed as for the miRNA data. MiRNAs and mRNAs were considered to be significantly deregulated if they showed an absolute $log_2$ fold-change of more than 1 and an adjusted p-value smaller than 0.05. They are considered to be deregulated if they show an absolute $log_2$ fold-change of more than 1 and an absolute Cohen's d value bigger than 0.5.

Top deregulated mRNA in Figs. 4–7 were determined based on the highest fold-change. Since the number of mRNA deregulated for each tissue was highly variable, we used the top 10% of deregulated mRNA for the tissues with more than 100 mRNA deregulated. Otherwise, all mRNA was taken for downstream pathway analysis.

We say the direction of deregulation matches if the sign of the two $log_2$ fold-changes is the same, so if both indicate upregulation or if both indicate downregulation.

### Pathway analysis using miEAA

GSEA using a miRNA list as input was performed using the miEAA tool[75] with standard parameters. The miRNA list is sorted by their log10 raw $p$ values calculated with the Wilcox Rank Sum Test and scaled with the direction of the deregulation, which means the sign of the $log_2$ fold change. A GSEA was then performed using the pathways from Gene Ontology that were derived via miRTarBase[76,77] and miRWalk[78]. $P$ values were adjusted using BH for each category independently.

For the smaller unordered miRNA sets (e.g., Fig. 3c), we performed Overrepresentation Analysis (ORA) using miEAA with standard parameters. The pathways were also derived for Gene Ontology via miRTarBase and miRWalk. $P$ values were adjusted using BH for each category independently.

ORA using the mRNA data as input was performed using the enrichGO function of clusterProfiler (version 4.6.0)[79]. The method uses either the GO Biological Process or all GO databases. We stated the method in the figure legends. $P$ values were adjusted using BH.

For all these pathway analysis methods, the pathways were considered to be significantly enriched or depleted if they showed an adj. p-value smaller than 0.05. Redundant GO terms were labeled using the minimal_set function of the OntologyIndex package (version 2.12)[80] and GO database org.Mm.eg.db (version 3.16.0) and removed from the list of pathways. Two pathways were considered to be related if one term is an ancestor of the other term. This was determined using the get_term_descendancy_matrix of OntologyIndex (version 2.12).

The pathways were clustered using GO_similarity and the binary_cut function from the simplifyEnrichment[81] package (version 1.8.0) with GO database org.Mm.eg.db (version 3.16.0). The pathways in each cluster were then summarized by counting the frequencies of each word in the set of pathway names and reporting the words that occur at least two times.

The results of the GSEA (genetrail[82])on the single-cell mRNA data were derived from GEO (Acc: GSE295428, see Methods).

### MiRNA-mRNA Interactions

MiRNA-mRNA target pairs were obtained from the TargetScan Release 8.0[12] for *M. musculus*. Only the pairs with a weighted context + + score about the 75th percentile were kept. As we expected the miRNA to play a role in deregulating mRNA expression, we filtered our miRNA-mRNA targets to only the mRNA targets of deregulated miRNA, which were themselves also deregulated as per the filtering conditions under the "Differential Expression Analysis" section.

### Overlap with circulating miRNAs

Average miRNA rpmm expression matrices for Mus musculus were downloaded from miRNATissueAtlas 2025[14]. Under the "tissue" biotype, we filtered for "blood", "plasma" and "serum". One of our selected miRNAs was determined to be present in the miRNATissueAtlas 2025 if it was detected above an average rpmm of at least 10 in "blood", "plasma" or "serum", respectively.

### Statistics and reproducibility

No statistical method was used to predetermine sample size. For the miRNA analyses, all available mice from each experimental group were included (see Samples and Cohorts). Samples were excluded during pre-processing based on pre-determined quality-control parameters. The experiments were not randomized, and the investigators were not blinded to group allocation during sample collection, processing, or outcome assessment. All statistical analyses were based on independent biological replicates. Technical replicates were used only during library preparation and plate processing to ensure uniform sample handling, but all statistical analyses were performed on independent biological replicates. All tests include information on sidedness (two-sided for correlations and Wilcoxon tests; one-sided for enrichment tests) and multiple-testing correction (Benjamini–Hochberg). All sample processing, library preparation, sequencing, and downstream analyses followed standardized protocols to ensure reproducibility.

### Reporting summary

Further information on research design is available in the Nature Portfolio Reporting Summary linked to this article.

## Data availability

The raw and processed miRNA data generated in this study have been deposited in the GEO database under accession code GSE294046. The raw and processed mRNA data generated in this study have been deposited in the GEO database under accession code GSE295428. Furthermore, the mRNA and miRNA data have been deposited in the NASE OSDR database under accessions OSD-904, OSD-905, OSD-906, OSD-907, OSD-908, OSD-909, OSD-910, OSD-911, OSD-912, OSD-913, OSD-914, OSD-915, OSD-916. Source data are provided with this paper.

## Code availability

The code used to create the figures (based on count data or the provided Supplementary Data) is available from GitHub[83].

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

## Acknowledgements

We thank all members of the Wyss-Coray and Meese labs, as well as the Keller lab, for their feedback and support. This study is partially funded by the Michael J. Fox Foundation and the Schaller-Nikolich Foundation. We acknowledge funding from the Deutsche Forschungsgemeinschaft (DFG, project number 469073465: Compute- und Storage Cluster). For the Development of the software pipeline used to process the single-cell data, we are supported by funds from the EU (Project 101057548-EPIVINF).

## Author contributions

F.G., S.R., A.E., and A.K. performed computational analysis and/or primary data processing. F.G., S.R., A.E., V.W., N.S., V.K., and A.K. wrote the manuscript with input from all authors. F.K., L.S., V.F., E.M., N.S., S.Q., T.W.-C. and A.K. organized the study. A.R., N.S., and K.C. were involved in sample collection. V.W., N.L., and M.K. performed the Sequencing preparation and the Sequencing.

## Funding

## Competing interests
The authors declare no competing interests.
