## [Transparent Peer Review file · Nature Communications]

MiRNAs shape mouse age-independent tissue adaptation to spaceflight via ECM and developmental pathways

Corresponding Author: Professor Andreas Keller

Version 0:

Reviewer comments:

Reviewer #1

(Remarks to the Author)

In this study, a collaborative team of highly experienced authors wished to map the impact of space travel on the levels of microRNAs (miRs) populations and their regulation of gene expression in diverse tissues of mice subjected to simulated space trip at two adult ages. To readily assess this impact, the authors might have performed global comparison of long and short RNA transcript profiles before and after space exposure; but rather than selecting this approach, they studied the impact of particular miRs by seeking their individual impact on specific predicted mRNA targets. This approach would yield some of the desired results, but in a partial manner dictated by the targeted analysis approach. It is the opinion of this reviewer that straight forward analysis of transcript and miR profiles could be more helpful in that it would yield a fuller picture, enabling to decide if miR changes suffice to induce the full impact picture or, alternatively, whether other regulators could be involved as well.

Reviewer #2

(Remarks to the Author)

This manuscript presents a highly valuable and comprehensive dataset investigating the effects of spaceflight on miRNA expression across 13 organs in mice, combined with an integrated mRNA analysis. The scope of the study, with 686 samples, is impressive and represents a significant resource for the space biology and aging research communities. The study offers important insights into miRNA-driven regulation of tissue remodeling, aging-related processes, and molecular adaptations to spaceflight stressors. However, there are substantial concerns that must be addressed to improve the rigor, clarity, and utility of this manuscript.

Major Comments:

1. Critical Concern: Brain Tissue Processing Ignores Spatial miRNA Heterogeneity

A significant limitation is the homogenization of whole brain tissue for RNA extraction without accounting for the well-documented regional specificity of miRNA expression in the brain. Pooling the entire brain can result in dilution or cancellation of region-specific miRNA signals, potentially masking biologically important effects. This is a critical flaw, especially given the central role the brain plays in spaceflight physiological adaptation.

-The authors should acknowledge this as a major limitation in both the Results and Discussion.

-At a minimum, they should validate the expression of key deregulated miRNAs in distinct brain regions using either existing archived samples (if available).

-Without addressing this, the conclusions regarding brain miRNA regulation during spaceflight are significantly weakened.

2. Circulating miRNAs should be addressed in the manuscript

The manuscript focuses entirely on tissue-specific miRNAs without considering circulating miRNAs, which are well-established as biomarkers of systemic physiological changes during spaceflight. Given the scale of this study, it is a missed opportunity not to analyze plasma, serum, or blood samples for key miRNAs.

-If blood-derived miRNA data were collected, it should be analyzed and reported. The minimal can be validation of the specific key miRNAs discussed in the paper in the circulation through ddPCR or other methods.

-If not, the authors should explicitly discuss this as a limitation and provide commentary on how tissue-specific miRNA changes may influence or be reflected in the circulating miRNA pool, which has implications for biomarker development and countermeasure strategies.

3. Lack of Experimental Validation of miRNA-mRNA Interactions

The manuscript relies entirely on computational predictions for miRNA-mRNA regulatory interactions without experimental validation. Given the central claim that miRNAs drive systemic and tissue-specific gene regulation under spaceflight, experimental validation (through qPCR) is essential.

- The authors should validate at least a subset of the most biologically relevant miRNA-mRNA pairs using methods such as qPCR for mRNA levels, miRNA mimic/inhibitor experiments, or luciferase reporter assays for direct binding confirmation.
- Suggested high-priority targets based on the manuscript include:
 - ITGB8, FLT1, RARB, MECOM, EYA4 — heavily targeted in SCAT, BAT, and GAT (linked to ECM, vasculature, and tissue remodeling).
 - PHF6, DGKE, STX6, AGO1, PHF6, MYB, RORB — involved in thymus, diaphragm, and brain regulatory circuits (related to DNA damage, immune function, and neurodevelopment).
 - Genes involved in mitochondrial or membrane remodeling pathways such as BSN, TSPAN4, and CRIM1.
- Special emphasis should be placed on validating miRNAs from the key families highlighted in the paper (MIR-17/92, MIR-1/133, MIR-15, MIR-8, and MIR-154) given their central role in the proposed regulatory architecture.

4. Ambiguity in Fold-Change Correlations (Figures 4C and 4D)

It is unclear whether the fold-change values used for the correlation analyses are based on miRNA data, mRNA data, or a combination of both. The authors must clearly state in the figure legends and main text which molecular data (miRNA or mRNA) underlies the fold-change correlations in Figures 4C and 4D.

5. Figure 4G — Missing Color Legend and Clarification

The color scheme used in Figure 4G is not explained. The meaning of different colors and shade gradients is unclear and prevents the reader from interpreting the data. A proper legend or detailed explanation should be added to clarify what the colors represent (e.g., tissue type, MIR family, directionality).

6. Figure 2D Visualization — Recommend UpSet Plot

The current visualization of overlapping deregulated miRNAs in Figure 2D is difficult to interpret. An UpSet plot would be a more effective visualization tool for depicting intersections and unique elements across multiple groups. This change would significantly improve the readability of the figure.

7. Lack of Clarity in Figure 3E Heatmap Description

The caption for Figure 3E does not adequately describe what the heatmap represents. Specifically, it should clearly state whether the heatmap displays log₂ fold changes, expression levels, or a miRNA-target interaction score. The associated dot plot also lacks an explanation. This figure requires improved annotation both in the caption and in the main text to ensure that readers can accurately interpret the key findings.

8. Data Must Be Deposited in NASA's Open Science Data Repository (OSDR)

This data should also be deposited in NASA's OSDR (<https://osdr.nasa.gov>) in addition to GEO. This is now standard practice for spaceflight related work and can provide a centralized and extremely valuable resource for the scientific community. This should include both the raw and processed data, metadata, and analysis files. Doing so will greatly improve the visibility, accessibility, and reuse of this important dataset by the broader research community.

Minor and Technical Comments:

1 Statistical Methods — Clarify Multiple Testing Correction

The manuscript mentions the use of fold-change and Cohen's D thresholds for differential miRNA expression but does not clearly state whether FDR or another multiple testing correction was applied. This should be clarified in the Methods.

2. Strain Differences Limit Aging Comparisons

The aging comparisons using the Tabula Muris Senis dataset are weakened by the use of different mouse strains (BALB/cAnNTac vs. C57BL/6JN). This should be explicitly discussed as a limitation, as strain-dependent miRNA expression is well-documented. Also the methods for this dataset is not provided in the methods section.

3. Radiation Effects Underrepresented in Discussion

The manuscript emphasizes microgravity but under-discusses radiation, despite miRNA signatures clearly associated with DNA damage responses. A deeper discussion on how space radiation contributes to miRNA dysregulation is warranted.

Reviewer #3

(Remarks to the Author)

This manuscript presented analysis of miRNA and mRNA expression in 686 mouse spaceflight samples derived from a variety of tissue types. While the scale of the dataset is notable, the manuscript lacks method details, particularly regarding how the spaceflight experiment was conducted and how the samples were collected/processed. Without this information, it is difficult to assess the quality and reliability of the data or the resulting analyses. Given the complexity of spaceflight experiments, such contextual information is essential.

The manuscript reported multiple differential expression analyses. However, the conclusions drawn from them are unclear. Moreover, there appears to be no experimental validation to support the findings, which raises concerns about the robustness of the descriptions.

Additionally, the GEO dataset referenced in the manuscript remains private at the time of review, preventing evaluation of the experimental design (referred to as the method description was lacking).

Given these significant limitations, I am unable to recommend this manuscript for publication in its current form.

Version 1:

Reviewer comments:

Reviewer #1

(Remarks to the Author)

The paper is acceptable now in its revised form.

Reviewer #2

(Remarks to the Author)

The authors have done a good job addressing my comments and I have no further comments.

Reviewer #3

(Remarks to the Author)

The manuscript was substantially improved in this revision. In the previous version, it was difficult to understand which datasets the authors used for the analysis, and even whether certain analyses reflected newly generated data or re-analyses of existing resources. The revised manuscript now clearly described the data sources, mission contexts, and analysis methods, allowing readers to understand the experimental design and the authors' interpretation. The authors demonstrated that seeking unifying small-RNA features across many tissue types naturally leads to inconclusive global patterns, and clarifying the constraints of such comparisons. Overall, the authors provide a valuable and well-curated resource that will meaningfully support future studies of spaceflight biology and small-RNA regulation. The manuscript is now clearly written, scientifically sound, and of high value to the field, and it is recommended for acceptance in its present form.

REVIEWER COMMENTS

Reviewer #1 (Remarks to the Author):

1. In this study, a collaborative team of highly experienced authors wished to map the impact of space travel on the levels of microRNAs (miRs) populations and their regulation of gene expression in diverse tissues of mice subjected to simulated space trip at two adult ages. To readily assess this impact, the authors might have performed global comparison of long and short RNA transcript profiles before and after space exposure; but rather than selecting this approach, they studied the impact of particular miRs by seeking their individual impact on specific predicted mRNA targets. This approach would yield some of the desired results, but in a partial manner dictated by the targeted analysis approach. It is the opinion of this reviewer that straight forward analysis of transcript and miR profiles could be more helpful in that it would yield a fuller picture, enabling to decide if miR changes suffice to induce the full impact picture or, alternatively, whether other regulators could be involved as well.

Background: The corresponding mRNA dataset was analyzed in detail and submitted as a separate manuscript. This is why the present work focuses on the miRNA dimension and does not repeat an extensive mRNA-level analysis here.

We agree that starting directly with targeted miRNA–mRNA pairing could limit interpretability. To improve clarity, we reorganized the presentation of the results so that the manuscript now first introduces all captured RNA classes, then focuses on miRNAs, and finally integrates the miRNAs with their mRNA targets. The section headers were updated accordingly:

1. “Spaceflight causes organ-specific murine small non-coding RNA expression changes”,
2. “Environment and spaceflight affect different tissues in murine miRNA profiles”,
3. “Spaceflight causes systemic and tissue-specific miRNA deregulation in gene expression” and
4. “Spaceflight causes both age-dependent and age-independent miRNA-mRNA regulation”.

To address the reviewer’s second point regarding additional non-coding RNA classes, including long non-coding RNAs and other short RNA species, we expanded the description of Fig. 1f to highlight spaceflight-related changes across these RNA types. Specifically, we now state: “Given this variation in RNA class distribution, we

analyzed how much of the observed expression variance could be attributed to spaceflight. The RNA types that were most affected by spaceflight were piRNA fragments (median: 2.35), lncRNA fragments (median: 1.65) and tRNA fragments (median: 1.95). This confirms that several RNA classes respond to spaceflight, which supports previous findings about their role in spaceflight adaptation and astronauts health^{23,24}. For miRNAs, spaceflight effects accounted for a median of 1.85% of the variance in expression across tissues (**Figure 1f**).".

While the other RNA classes may also contribute to spaceflight adaptations, the small-RNA sequencing protocol used here is optimized for miRNA detection, making miRNAs the most robust and interpretable component of our dataset. For this reason, we focused our downstream analyses on miRNAs and their mRNA targets. Nevertheless, to ensure transparency and facilitate further research, the complete dataset, including all detected RNA classes and their DE analysis results, is publicly available. In addition, although a complementary single-cell RNA-seq dataset was generated as part of the same mission, a detailed analysis of these data is beyond the scope of the present manuscript and is part of a separate study focused on cellular composition and transcriptional responses across tissues. Here, we use the single-cell data only in a limited, supportive manner to maintain a clear focus on miRNA-mediated regulation.

Reviewer #2 (Remarks to the Author):

This manuscript presents a highly valuable and comprehensive dataset investigating the effects of spaceflight on miRNA expression across 13 organs in mice, combined with an integrated mRNA analysis. The scope of the study, with 686 samples, is impressive and represents a significant resource for the space biology and aging research communities. The study offers important insights into miRNA-driven regulation of tissue remodeling, aging-related processes, and molecular adaptations to spaceflight stressors. However, there are substantial concerns that must be addressed to improve the rigor, clarity, and utility of this manuscript.

We thank the reviewer for the very constructive feedback and the overall positive assessment.

1. **Critical Concern: Brain Tissue Processing Ignores Spatial miRNA Heterogeneity**
A significant limitation is the homogenization of whole brain tissue for RNA extraction without accounting for the well-documented regional specificity of miRNA expression in the brain. Pooling the entire brain can result in dilution or cancellation of region-specific miRNA signals, potentially masking biologically important effects. This is a critical flaw, especially given the central role the brain plays in spaceflight physiological adaptation.
 - The authors should acknowledge this as a major limitation in both the Results and Discussion.
 - At a minimum, they should validate the expression of key deregulated miRNAs in distinct brain regions using either existing archived samples (if available).
 - Without addressing this, the conclusions regarding brain miRNA regulation during spaceflight are significantly weakened.

We agree that regional heterogeneity in brain miRNA expression is well documented and that dissecting region-specific effects would further enhance the interpretation of our findings. As archived region-specific brain samples are not available from this mission, which precludes additional validation experiments. To address this, we have now expanded the Discussion to explicitly acknowledge this limitation and to reference comparative datasets demonstrating brain region specificity in miRNA expression in humans ¹ and in mouse aging studies from other strains ². In our dataset, mmu-miR-20b-5p was the only brain-associated miRNA identified. Its human homolog has shown region-specific expression ¹, whereas in

our mouse brain aging atlas it did not appear region-specific, likely due to strain differences (BALB/cAnNTac vs. C57BL/6JN).

We have revised the manuscript to include the following clarification in the limitation section:

“Taking the brain as a whole during library preparation is likely to remove brain region specific effects of spaceflight, as several miRNA are known to show localized expression ². While our highlighted miRNA mmu-miR-20b-5p was not found to be brain-region specifically expressed, it is worth considering the known effect of strain on miRNA expression; as our previously published mouse brain aging atlas ² contains C57BL/6JN vs. BALB/cAnNTac used in this study. It might be that the deregulation observed here is resulting from the combined deregulation in multiple regions or in only a single region.”

2. Circulating miRNAs should be addressed in the manuscript
The manuscript focuses entirely on tissue-specific miRNAs without considering circulating miRNAs, which are well-established as biomarkers of systemic physiological changes during spaceflight. Given the scale of this study, it is a missed opportunity not to analyze plasma, serum, or blood samples for key miRNAs.
 - If blood-derived miRNA data were collected, it should be analyzed and reported. The minimal can be validation of the specific key miRNAs discussed in the paper in the circulation through ddPCR or other methods.
 - If not, the authors should explicitly discuss this as a limitation and provide commentary on how tissue-specific miRNA changes may influence or be reflected in the circulating miRNA pool, which has implications for biomarker development and countermeasure strategies.

We agree on this important point. Circulating miRNAs would indeed provide valuable information to connect tissue-specific effects with systemic regulation. Unfortunately, blood or plasma samples were not collected during tissue harvesting for this mission, preventing direct measurement of circulating miRNAs.

To nevertheless address the reviewer’s concern, we took two complementary steps:

- We compared our findings to previously reported circulating miRNAs affected by spaceflight. As noted in the manuscript, only one miRNA (miR-17-5p) overlaps with those reported by Malkani et al. (2020)³. This limited overlap likely reflects differences in mouse strain (BALB/cAnNTac vs. C57BL/6J) and the well-

documented divergence between circulating and solid-tissue miRNA signatures^{4,5}.

- We assessed whether the miRNAs we identified as spaceflight-responsive are known to circulate. Using the miRNA TissueAtlas 2025 (mean expression >10 rpmm in blood, plasma, or serum), we found that 49 of the 56 miRNAs highlighted in Figures 2h, 3c, and 4h are detected in at least one circulating medium. This indicates that many of the affected miRNAs have the potential to be reflected in circulation and may therefore possess systemic biomarker relevance.

We now explicitly discuss this in the manuscript:

“[...] Similarly, circulating miRNAs could provide a systemic view of the tissue-specific changes observed. Although circulating miRNAs were not collected in this study, comparison to published spaceflight datasets and reference expression profiles indicates that many of the tissue miRNAs affected here (49 out of 56) are detectable in blood, plasma, or serum, suggesting potential systemic biomarker relevance (Supplementary Table 9).”

3. Lack of Experimental Validation of miRNA-mRNA Interactions

The manuscript relies entirely on computational predictions for miRNA-mRNA regulatory interactions without experimental validation. Given the central claim that miRNAs drive systemic and tissue-specific gene regulation under spaceflight, experimental validation (through qPCR) is essential.

- The authors should validate at least a subset of the most biologically relevant miRNA-mRNA pairs using methods such as qPCR for mRNA levels, miRNA mimic/inhibitor experiments, or luciferase reporter assays for direct binding confirmation.
- Suggested high-priority targets based on the manuscript include:
 - ITGB8, FLT1, RARB, MECOM, EYA4 — heavily targeted in SCAT, BAT, and GAT (linked to ECM, vasculature, and tissue remodeling).
 - PHF6, DGKE, STX6, AGO1, PHF6, MYB, RORB — involved in thymus, diaphragm, and brain regulatory circuits (related to DNA damage, immune function, and neurodevelopment).
 - Genes involved in mitochondrial or membrane remodeling pathways such as BSN, TSPAN4, and CRIM1.
 - Special emphasis should be placed on validating miRNAs from the key families highlighted in the paper (MIR-17/92, MIR-1/133, MIR-15, MIR-8, and MIR-154) given their central role in the proposed regulatory architecture.

We thank the reviewer for this important comment and fully agree that validation of key miRNA–mRNA interactions is essential. Our group routinely performs such validations (e.g., Kern et al.⁶, Hart et al.⁷, Diener et al.⁸). However, in the context of this spaceflight mission, no biological material remained available, and the unique nature of the experiment prevents generating new samples. Since numerous validated targets for the identified miRNAs are already documented in established databases, additional validation of just a few new targets would, in our opinion, not have been proportionate. Instead, we leveraged these existing resources to place our findings in context.

To address this limitation, we performed a cross-reference analysis of all deregulated miRNA–mRNA pairs against the experimentally validated interactions listed in TarBase 9 and miRTarBase 2025. For spaceflight-related deregulated pairs, 9,569 out of 62,754 (15%) were experimentally validated, and for age-related pairs, 253 out of 1,516 (16%) were validated. Notably, experimentally supported interactions include CRIM1, DGKE, FLT1, MECOM, TSPAN4, MYB, and RARB, several of which were highlighted by the reviewer.

We also repeated key analyses using only validated miRNA–mRNA pairs and observed consistent pathway-level enrichment, confirming the robustness of our main conclusions.

We now state this in the manuscript:

“Since the miRNA-mRNA targeting can be spurious, we further compared our deregulated miRNA-mRNA pairing against Tarbase 9⁹ and miRTarbase 2025¹⁰ to see if they have been validated in other studies. Out of 62,754 targets we highlighted, 9,569 have previously been experimentally validated in the literature (Supplementary Table 6). Furthermore, a sub-analysis using just the validated miRNA-mRNA pairs confirmed that our main results remain consistent (Supplementary Figure 6).”

“We also compare the miRNA-mRNA targeting we highlight here against Tarbase 9 and miRTarBase 2025, confirming that our conclusion remains consistent even when using only experimentally validated targeting (Supplementary Figure 9, Supplementary Table 8).”

“We have attempted to partially validate the miRNA-mRNA interactions by comparison against experimentally validated data in publicly available websites to see if our conclusions change. Due to the incomplete nature of miRNA-mRNA experimental validation data and the existence of false positives in predicted targets, we were able to verify a fraction of the targeting. Interestingly, our final conclusion remains unchanged.

4. Ambiguity in Fold-Change Correlations (Figures 4C and 4D)
It is unclear whether the fold-change values used for the correlation analyses are based on miRNA data, mRNA data, or a combination of both. The authors must clearly state in the figure legends and main text which molecular data (miRNA or mRNA) underlies the fold-change correlations in Figures 4C and 4D.

We thank the reviewer for pointing out this ambiguity. We have clarified in the figure legends and main text that Figures 4C–D depict miRNA fold-change correlations. The revised captions now read:

Figure 4c: “Correlation between miRNA fold-changes of FL vs. HGC across the age groups 3 Month, 8 Month and All Ages. Significant correlations (adj. P-value < 0.05, abs. Pearson’s correlation >0.5, BH adjustment) are shown with a dot.”

Figure 4d: “Effects of spaceflight on the different age groups. SCAT and Liver correspond to examples of tissues with the highest correlation between miRNA fold-changes (Pearson’s R value = 0.65 and 0.62), while MAT and diaphragm correspond to tissues with the lowest correlation (Pearson’s R value of = -0.29 and -0.40).”

5. Figure 4G — Missing Color Legend and Clarification
The color scheme used in Figure 4G is not explained. The meaning of different colors and shade gradients is unclear and prevents the reader from interpreting the data. A proper legend or detailed explanation should be added to clarify what the colors represent (e.g., tissue type, MIR family, directionality).

We have clarified the color coding in the figure legend as follows: “Number of miRNAs from (e) for each tissue grouped by the MIR Family of origin, for all MIR-families with more than 2 miRNAs in one tissue. Dot color corresponds to the tissue color while FL 3M and FL 8M groups are in yellow and violet.”

6. Figure 2D Visualization — Recommend UpSet Plot
The current visualization of overlapping deregulated miRNAs in Figure 2D is difficult to interpret. An UpSet plot would be a more effective visualization tool for depicting intersections and unique elements across multiple groups. This change would significantly improve the readability of the figure.
We thank the reviewer for this suggestion. We generated UpSet plots for all 13 tissues. These are now included as Supplementary Figure 2.
For the main figure (previous Figure 2D, now 2G), we kept barplots to summarize the overlaps that are most relevant. We think this strikes a good balance: it keeps the

main figure easy to read while still providing the full UpSet visualizations in the Supplementary material for anyone who wants to explore all intersections.

7. Lack of Clarity in Figure 3E Heatmap Description
The caption for Figure 3E does not adequately describe what the heatmap represents. Specifically, it should clearly state whether the heatmap displays log₂ fold changes, expression levels, or a miRNA-target interaction score. The associated dot plot also lacks an explanation. This figure requires improved annotation both in the caption and in the main text to ensure that readers can accurately interpret the key findings.

We have revised Figure 3E and its caption for clarity: “Heatmap of average log₂ FC between FL and HC for each Tissue for mRNA most heavily targetted by miRNA (left) along with the miRNAs targetting the mRNA (right). A dot represents targeting, while no dot means no targeting. Color of the dot represents the Tissue where the targeting is happening. The miRNAs are grouped based on which MIR family they belong to. Stacked barplots on the right show the count of the number of miRNAs targeting each mRNA in each tissue.”

8. Data Must Be Deposited in NASA’s Open Science Data Repository (OSDR)
This data should also be deposited in NASA’s OSDR (<https://osdr.nasa.gov>) in addition to GEO. This is now standard practice for spaceflight related work and can provide a centralized and extremely valuable resource for the scientific community. This should include both the raw and processed data, metadata, and analysis files. Doing so will greatly improve the visibility, accessibility, and reuse of this important dataset by the broader research community.

We thank the reviewer for this suggestion. We are already in contact with the NASA OSDR curation team. Due to the dataset size (~2.3 TB), the raw and processed data uploaded to GEO/SRA will be mirrored to OSDR by NASA’s data curation team at the time of publication, ensuring full accessibility.

Minor and Technical Comments:

1. Statistical Methods — Clarify Multiple Testing Correction
The manuscript mentions the use of fold-change and Cohen's D thresholds for differential miRNA expression but does not clearly state whether FDR or another multiple testing correction was applied. This should be clarified in the Methods.

We apologize for the confusion about this point. We prioritized the differential expression results based on fold-change and Cohen's D effect size rather than on p-value ranking. Nevertheless, all p-values were calculated and adjusted using the Benjamini–Hochberg (BH) method, as described in the Methods section under Differential Expression Analysis:

“Differential expression analysis of the miRNAs was performed using the filtered rpmm values with a Wilcox Rank Sum Test and p-values were adjusted using Benjamini Hochberg (BH).”

For transparency, both raw and adjusted p-values are provided in Supplementary Table 2 for readers who wish to explore them.

2. Strain Differences Limit Aging Comparisons
The aging comparisons using the Tabula Muris Senis dataset are weakened by the use of different mouse strains (BALB/cAnNTac vs. C57BL/6JN). This should be explicitly discussed as a limitation, as strain-dependent miRNA expression is well-documented. Also the methods for this dataset is not provided in the methods section.

We agree and have emphasized this limitation in the Discussion: “Notably, the *M. musculus* strain for our spaceflight study (BALB/cAnNTac) differed from TMS (vC57BL/6JN). As miRNA expression is not only tissue and age- but also strain-specific, this fact adds noise to our comparison.”

We have also clarified in the Methods that both datasets were processed using identical analytical pipelines:

“All miRNA datasets, for Spaceflight and Tabula Muris Senis was processed in identical manner.”

3. Radiation Effects Underrepresented in Discussion
The manuscript emphasizes microgravity but under-discusses radiation, despite miRNA signatures clearly associated with DNA damage responses. A deeper discussion on how space radiation contributes to miRNA dysregulation is warranted.

We agree with the reviewer that radiation-induced DNA damage plays a role and that we observe this in our data, as mentioned in the Discussion: “For DNA damage, we saw strong signals of nuclear environment remodeling, likely as a response to radiation exposure. Members of the miR-17/92 cluster such as miR-92b and miR-363-3p⁴² are involved in radiation exposure-related DNA damage control. Additionally, LET-7, which influences both systemic and more tissue-specific pathways, plays a well-documented role in radiation response, including exposure to deep-space radiation^{16,43}.” However, we observe that this effect is somewhat smaller in comparison to ECM remodelling. DNA damage signals are found most noticeably in the Thymus and GAT while ECM remodelling occurs extensively across most of the sampled tissues. To address this, we add the following line to the Discussion: “Interestingly, the limited observation of DNA damage response is somewhat counter to what is seen in circulating miRNAs, where this response dominates. While this might be partially due to the different strains of mice used in the study, it might also hint at differing response times of blood and solid tissue to the ionizing radiation prevalent on the ISS, as solid tissue is known to have slower response kinetics in comparison to blood¹¹. However, due to the absence of blood samples, this hypothesis cannot be verified from our data directly.”

Reviewer #3 (Remarks to the Author):

1. This manuscript presented analysis of miRNA and mRNA expression in 686 mouse spaceflight samples derived from a variety of tissue types. While the scale of the dataset is notable, the manuscript lacks method details, particularly regarding how the spaceflight experiment was conducted and how the samples were collected/processed. Without this information, it is difficult to assess the quality and reliability of the data or the resulting analyses. Given the complexity of spaceflight experiments, such contextual information is essential.

We thank the reviewer for their feedback. In response, we have expanded the Methods section to include detailed descriptions of the spaceflight experiment and sample collection, as well as direct references to the official NASA protocols used for this mission. The following text has been revised/added: “VGC mice were housed at NASA Kennedy Space Center in standard vivarium conditions at 4 mice per cage, 20-22.2 °C, 12hr light 12hr dark. HGC mice housing was matched to the conditions experienced by Flight mice (as closely as possible) by using double density housing (the same Rodent Habitats (AEM-X) and Animal Enclosure Module Transporters (AEM-T) as the flight animals), with identical food, matched temperature, humidity, and carbon dioxide levels based on actual flight environmental monitoring. Each habitat contained ten mice (five per side) and included enrichment huts as part of the AEM-X configuration. To dissect all mice together in matching conditions, VGC and HGC mice were transported to Scripps Institute just before the return of the Flight mice and were dissected in matching conditions (+/- 1 day).”

and

“All tissue collection and handling procedures were performed under NASA IACUC protocol #FLT-18-116 and Scripps protocol #09-0004, following the Rodent Research Reference Mission 1 (RR-8) operational guidelines. Further details about Rodent Research Reference Mission 1 can be found at (<https://osdr.nasa.gov/bio/repo/data/payloads/RRRM-1%20%28RR-8%29>).”

We now also reference the publicly available NASA Open Science Data Repository protocol for this mission in the Methods section.

2. The manuscript reported multiple differential expression analyses. However, the conclusions drawn from them are unclear. Moreover, there appears to be no

experimental validation to support the findings, which raises concerns about the robustness of the descriptions.

We appreciate the reviewer's careful assessment. We have clarified in the revised manuscript that our conclusions are not based on individual differential expression results but rather on the multimodal analysis of miRNA and mRNA. Each miRNA and mRNA deregulation serves as an intermediate step in the analysis process. The conclusions from these show the impact of multiple miRNA families on pathways related to ECM remodelling, development pathways and DNA damage and are discussed extensively in the Discussions section, with a concise summary highlighting the main biological message. We acknowledge the lack of experimental validation as a major limitation of this study. To strengthen confidence in the findings, we now include supplementary tables cross-referencing our predicted miRNA–mRNA interactions with validated entries from TarBase and miRTarBase (Supplementary Tables 6 and 8, Supplementary Figures 5 and 8). While further experimental validation under true microgravity conditions is logistically challenging, we agree that follow-up experiments in simulated microgravity systems on Earth would provide valuable confirmation. We now discuss specific candidate approaches in the revised Discussion, referencing prior studies that have observed similar ECM and developmental pathway effects under simulated microgravity^{12,13}

3. Additionally, the GEO dataset referenced in the manuscript remains private at the time of review, preventing evaluation of the experimental design (referred as the method description was lacking).

We thank the reviewer for noting this. The dataset will be made fully available upon publication. A reviewer access token has been provided with this revision to enable data inspection during the review process.

4. Given these significant limitations, I am unable to recommend this manuscript for publication in its current form.

We appreciate the reviewer's feedback, which has helped us substantially improve the clarity and transparency of our study. We believe that the expanded Methods section, inclusion of validated target references, and improved data accessibility address the reviewer's main concerns.

References

- 1 Wang, R. *et al.* Human microRNA (miR-20b-5p) modulates Alzheimer's disease pathways and neuronal function, and a specific polymorphism close to the MIR20B gene influences Alzheimer's biomarkers. *Molecular Psychiatry* **27**, 1256-1273 (2022). <https://doi.org:10.1038/s41380-021-01351-3>
- 2 Engel, A. *et al.* A spatio-temporal brain miRNA expression atlas identifies sex-independent age-related microglial driven miR-155-5p increase. *Nature Communications* **16**, 4588 (2025). <https://doi.org:10.1038/s41467-025-59860-6>
- 3 Malkani, S. *et al.* Circulating miRNA Spaceflight Signature Reveals Targets for Countermeasure Development. *Cell Reports* **33**, 108448 (2020). <https://doi.org:https://doi.org/10.1016/j.celrep.2020.108448>
- 4 Shah, R. *et al.* Discordant Expression of Circulating microRNA from Cellular and Extracellular Sources.
- 5 Pigati, L. *et al.* Selective Release of MicroRNA Species from Normal and Malignant Mammary Epithelial Cells. *PLOS ONE* **5**, e13515 (2010). <https://doi.org:10.1371/journal.pone.0013515>
- 6 Kern, F. *et al.* Validation of human microRNA target pathways enables evaluation of target prediction tools. *Nucleic Acids Res* **49**, 127-144 (2021). <https://doi.org:10.1093/nar/gkaa1161>
- 7 Hart, M. *et al.* The deterministic role of 5-mers in microRNA-gene targeting. *RNA Biol* **15**, 819-825 (2018). <https://doi.org:10.1080/15476286.2018.1462652>
- 8 Diener, C. *et al.* Quantitative and time-resolved miRNA pattern of early human T cell activation. *Nucleic Acids Res* **48**, 10164-10183 (2020). <https://doi.org:10.1093/nar/gkaa788>
- 9 Skoufos, G. *et al.* TarBase-v9.0 extends experimentally supported miRNA-gene interactions to cell-types and virally encoded miRNAs. *Nucleic Acids Research* **52**, D304-D310 (2024). <https://doi.org:10.1093/nar/gkad1071>
- 10 Cui, S. *et al.* miRTarBase 2025: updates to the collection of experimentally validated microRNA-target interactions. *Nucleic Acids Res* **53**, D147-d156 (2025). <https://doi.org:10.1093/nar/gkae1072>
- 11 Stewart-Ornstein, J. *et al.* p53 dynamics vary between tissues and are linked with radiation sensitivity. *Nature Communications* **12**, 898 (2021). <https://doi.org:10.1038/s41467-021-21145-z>
- 12 Benavides Damm, T., Walther, I., Wüest, S. L., Sekler, J. & Egli, M. Cell cultivation under different gravitational loads using a novel random positioning incubator. *Biotechnology and Bioengineering* **111**, 1180-1190 (2014). <https://doi.org:https://doi.org/10.1002/bit.25179>
- 13 Arfat, Y. *et al.* Physiological Effects of Microgravity on Bone Cells. *Calcified Tissue International* **94**, 569-579 (2014). <https://doi.org:10.1007/s00223-014-9851-x>